# Context memory formed in medial prefrontal cortex during infancy enhances learning in adulthood

María P. Contreras [1,2,6], Marta Mendez [3], Xia Shan[1,2], Julia Fechner [1,2], Anuck Sawangjit [1], Jan Born [1,4,5,7] ✉ & Marion Inostroza [1,7] ✉

Adult behavior is commonly thought to be shaped by early-life experience, although episodes experienced during infancy appear to be forgotten. Exposing male rats during infancy to discrete spatial experience we show that these rats in adulthood are significantly better at forming a spatial memory than control rats without such infantile experience. We moreover show that the adult rats' improved spatial memory capability is mainly based on memory for context information during the infantile experiences. Infantile spatial experience increased c-Fos activity at memory testing during adulthood in the prelimbic medial prefrontal cortex (mPFC), but not in the hippocampus. Inhibiting prelimbic mPFC at testing during adulthood abolished the enhancing effect of infantile spatial experience on learning. Adult spatial memory capability only benefitted from spatial experience occurring during the sensitive period of infancy, but not when occurring later during childhood, and when sleep followed the infantile experience. In conclusion, the infantile brain, by a sleep-dependent mechanism, favors consolidation of memory for the context in which episodes are experienced. These representations comprise mPFC regions and context-dependently facilitate learning in adulthood.

Early life experience critically forms behaviour in adulthood. This is a long-standing and prominent cultural idea that has been at the core of modern psychology as well as recent research linking, for example, traumatic experience during early life with capabilities to cope with stress in adulthood[1,2]. The importance of early experience for adult behaviour, however, appears to contrast with the observation that the memories formed of the multiple episodes during infancy are altogether rapidly forgotten, a phenomenon sometimes referred to as infantile amnesia[3,4]. Yet, there is also evidence suggesting that such infantile memories after they were forgotten, can be reinstated later in life by presenting appropriate reminders[5–7]. A conceptual proposal

reconciling these apparently diverging observations is, that rather than being forgotten, episodic memories formed of experience during infancy are transformed into more abstract memories containing only the gist of the experiences. Such transformed memories, then, serve as enduring supraordinate knowledge to facilitate behavioural adaptation during adulthood[8–12].

Despite the outstanding theoretical interest, little experimental work has been performed on how a certain separate episodic experience during infancy is consolidated into memory to feed into adult knowledge systems[11,13–18]. Moreover, most of this research employed highly aversive stimuli that, due to their stressful nature, invoke

[1]Institute of Medical Psychology and Behavioral Neurobiology, University of Tübingen, Tübingen, Germany. [2]Graduate School of Neural & Behavioral Science, International Max Planck Research School, Tübingen, Germany. [3]Laboratory of Neuroscience, Department of Psychology, Instituto de Neurociencias del Principado de Asturias (INEUROPA), University of Oviedo, Plaza Feijoo, Oviedo, Spain. [4]Werner Reichert Center for Integrative Neuroscience, University of Tübingen, Tübingen, Germany. [5]German Center for Diabetes Research (DZD)—Institute for Diabetes Research and Metabolic Diseases of the Helmholtz Center Munich (IDM) at the University Tübingen, Tübingen, Germany. [6]Present address: Leibniz-Institute of Neurobiology, Magdeburg, Germany. [7]These authors contributed equally: Jan Born, Marion Inostroza. ✉e-mail: jan.born@uni-tuebingen.de; marion.inostroza@uni-tuebingen.de

memory processes distinctly differing from more neutral everyday experience[5,19,20], and the very few studies examining effects of non-aversive experience all employed non-specific stimulation covering extended periods of postnatal life (like prolonged exposure to novel environments[21–23]). Against this backdrop, in the present study we adopted a novel experimental approach to tackle the question how discrete non-emotional spatial experience during infancy becomes integrated into persisting knowledge systems and eventually impacts adult behaviour. We took advantage of the well-controlled conditions of a rat model to show that such discrete experiences, i.e., the exposure to a change in the spatial configuration of two objects in an experimental arena for only short 5 min intervals on 4 days during a rat's infancy, distinctly enhances the rat's spatial learning ability during adulthood. To induce spatial experience during infancy, we employed a simple procedure: rats of a Spatial-experience group were placed in an arena with two identical objects for 5 min. Then, after a 5 min break, the rats re-entered the arena for another 5 min; however, this time one of the objects was moved to another location (Fig. 1A). A control group of rats was exposed to Object-experience during infancy by only changing the kind of one of the two objects between the two 5 min

exposure periods, instead of its spatial configuration. An additional No-experience control group did not undergo experimental arena visits during infancy. During adulthood, around postnatal day (PD) 80, all rats were tested on a classical object-place recognition (OPR) task, with a 3 h delay between encoding and retrieval testing, to assess the animal's capability to form stable spatial representations.

## Results

### Infantile spatial experience enhances OPR performance at adulthood

At testing in adulthood, the rats with infantile spatial experience showed enhanced capabilities to form spatial memory on the OPR task. Retrieval performance of this group was significantly higher than in both the Object-experience and the No-experience control groups ($F(4.944, 121.135) = 2.989$, $P = 0.014$ for Group × Minute ANOVA interaction). The enhancement was largest during the first minute of the retrieval phase which is typically most sensitive to the memory effect[24] ($F(2, 51) = 4.464$, $P = 0.017$, for main effect of Group, see Fig. 1B for results from pairwise statistical comparisons). In fact, during this first minute of adulthood OPR testing only the Spatial-experience group

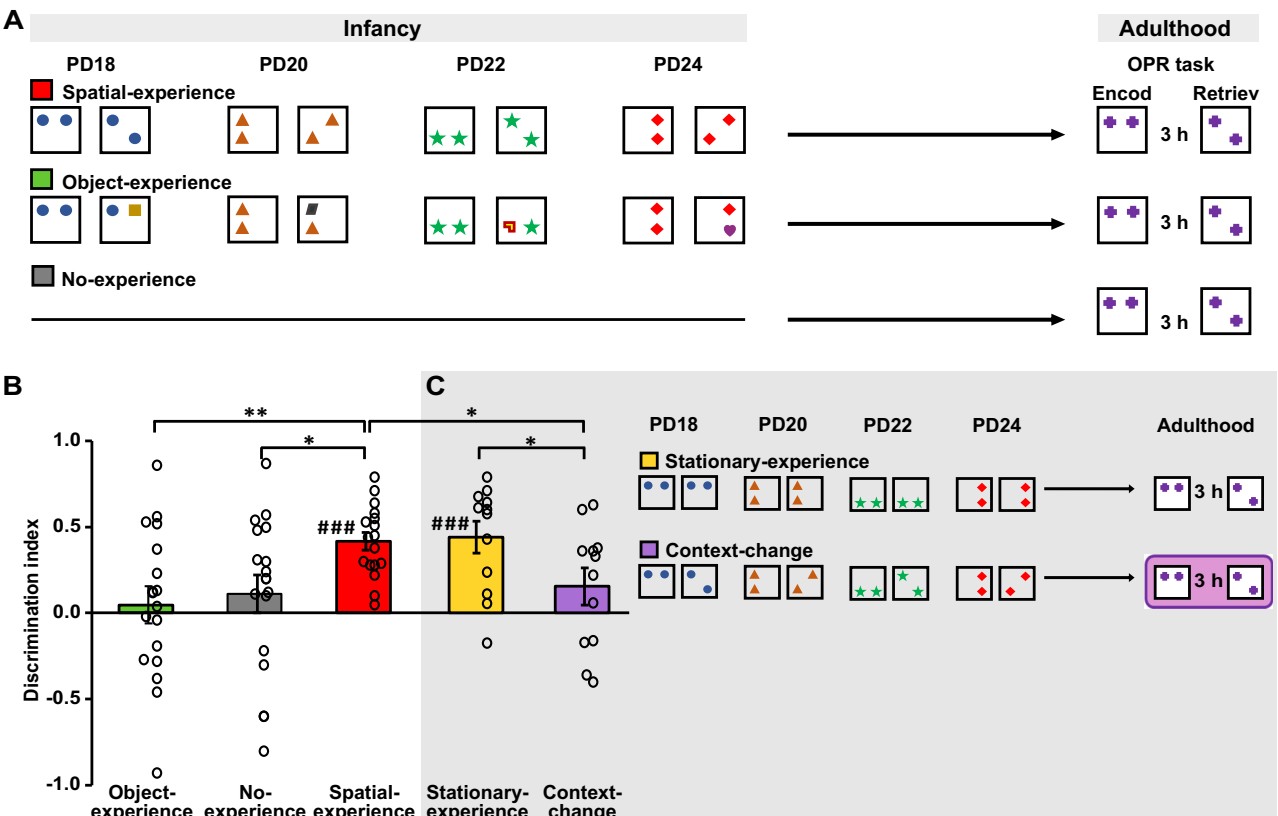

**Fig. 1 | Effect of infantile spatial experience on adult OPR performance.**
**A** General procedure: During infancy, pups of the Spatial-experience group ($n = 17$, red) were placed in an arena with two identical objects for 5 min and, after a 5 min break, re-entered to the arena but this time, one of the objects was displaced to a new location. Different spatial configurations and objects were used at the four exposures on PD18, 20, 22, and 24. For the Object-experience group ($n = 17$ rats, green), instead of a change in object location, one of the objects was replaced by another, in the second 5 min period. The No-experience group ($n = 18$ rats, grey) had no arena visits during infancy. At adulthood (~PD80), all groups were tested on a classical object-place recognition (OPR) task with a 3 h delay between encoding and retrieval testing. **B** OPR memory (mean ± SEM discrimination ratios during 1st minute of retrieval phase, dot plots overlaid) at adulthood testing. Only rats with spatial experience during infancy displayed significant OPR memory ###P = 0.000 for one-sample t-test (two-sided) against chance level *P = 0.023 and **P = 0.004 for pairwise comparisons (two-sided t-test) between experimental groups. **C** Grey

shaded shows the OPR memory performance (mean ± SEM discrimination ratios during 1st minute of retrieval phase, dot plots overlaid) at adulthood and the procedure of additional control experiments (right). For the rats of the Stationary-experience group ($n = 12$ rats, yellow) both the objects and their spatial configuration remained unchanged at the two visits of each infantile exposure. Procedures for the Context-change group ($n = 12$ rats, purple) were the same as for the Spatial-experience group, except that OPR testing at adulthood was performed in an entirely different context. Whereas the Stationary-experience group showed the same enhanced OPR performance as the Spatial-experience group, the Context-change groups did not show significant OPR memory (bottom). ###P = 0.000 for one-sample t-test (two-sided) against chance level; *P < 0.05 for pairwise comparisons (two-sided t-test) between experimental groups. (see Figs. S1, S4 for discrimination ratios for entire 5-min retrieval phase). Source data are provided as a Source Data file.

exhibited consistent preference for the displaced object expressing it throughout the 5 min retrieval phase ($t(16)$ = 8.323, 5.711, 4.326, 3.579 and 3.894, all $P < 0.003$ for comparisons with chance level performance per minute, Fig. S1). The control rats with Object-experience during infancy did not express any significant spatial memory throughout the test period (all $t(16) > 0.423$ and $P > 0.074$), and the No-experience group only transiently expressed memory (2nd and 3rd min of retrieval phase; $t(17)$ = 2.147 and 2.456, $P < 0.046$). The result for the No-experience group replicates previous studies testing OPR memory in adult rats using the same long 3 h delay[25], and show that naïve rats do not form an OPR memory as robust as that in the Spatial-experience group, expressing itself already in the 1st min of retrieval testing. Analysis of behavioural control parameters revealed that Spatial-experience and Object-experience groups were closely comparable with respect to total exploration time (Fig. S2A). However, rats of the Spatial-experience group travelled a slightly greater distance than the rats of the Object-experience group ($t(32) = -2.055$, $P = 0.048$), possibly reflecting general arousing effects on locomotion resulting from stimulation specifically of spatial systems during infancy. We excluded travelled distance as a factor possibly confounding memory performance in additional analyses including distance travelled as covariate, which confirmed significance for the difference in discrimination indexes between the Spatial- and Object-experience groups ($F(1, 31) = 8.416$, $P = 0.007$). During the encoding phase, total object exploration and distance travelled were decreased in the No-experience group ($t(33) > 3.6$, $P > 0.001$, for comparisons with the two other groups). Control analyses using total exploration and distance travelled at encoding as covariates did not provide any hint that these parameters biased the observed differences in the discrimination index between groups ($P > 0.153$ for the covariates, Fig. S2).

Control analyses of the behaviour at the four exposures during infancy indicated that the pups' interest towards the objects and arena environment at these exposures was comparable between the Spatial-experience and Object-experience groups (Fig. S3). Overall, these results demonstrate that discrete and short-lasting spatial experience during infancy distinctly impacts the capability to form stable spatial representations in adulthood.

### Context information at infantile exposures is sufficient for enhancing the adult rat's OPR performance

The infantile exposures in our Spatial-experience group comprised changes in the configuration of the two objects (from the first to the second visit of the arena) with the environmental arena context remaining the same throughout the four exposures. The arena context for the Spatial-experience group being the same as for the Object-experience group suggests that the rats of the Spatial-experience group relied on the experienced changes in the spatial configuration of the two objects, rather than on contextual information, to enhance OPR performance at testing in adulthood. Yet, it could also be argued that in the Object-experience group the change in the object during the infantile exposures distracted these pups from encoding context information potentially relevant for their OPR performance during adulthood. Hence, to further specify the information the rats of the Spatial-experience group used at infantile exposures to enhance their adult OPR performance, we examined an additional group of rats (Stationary-experience). Rats of this group were subjected to basically the same procedures as the Object-experience group, except that the objects during the two visits of the arena remained the same (Fig. 1C). These exposures, thus, did not only lack any change in the spatial configuration of the two objects but also the potentially distracting effect of a change in the objects. Notably, at OPR testing during adulthood, the Stationary-experience group profited from the infantile exposures in the same way as the Spatial-experience group, i.e., OPR performance of these rats was significantly better than that of the

Object-experience and No-experience control groups ($F(2, 46) = 3.409$, $P = 0.042$, for the group main effect across these 3 groups), and closely comparable with that of the Spatial-experience group ($P = 0.816$, for the pairwise comparison between these groups). Similar to the Spatial-experience group, discrimination preference in the rats of the Stationary-experience group appeared to remain at a high level throughout the 5 min retrieval phase (Fig. S4). Indeed, these findings suggest that, for enhancing OPR performance at adulthood, the pups need to only maintain the spatial context information at the infantile exposures. (Note that encoding the context implicates an event experienced which in the Stationary-experience group was represented by the different objects in at each infantile exposure.) On the other hand, the exposure to a change in the spatial configuration of the two objects—as experienced by the pups of the Spatial-experience group —is not required.

### Effects of infantile spatial experience on the adult rat's learning capability are context-dependent

If the enhanced OPR performance at adulthood in rats of the Spatial-experience group was primarily owed to the memorization of environmental context information encoded during the exposures during infancy, we would expect that the enhancement in OPR performance in these rats is restricted to the same context as that during the infantile exposures. To test this hypothesis, we compared the Spatial-experience rats which performed the OPR task at adulthood in a very similar context as that used for inducing spatial experience during infancy (same experimenter, same distal cues) with a control group of Spatial-experience rats which performed the OPR task at adulthood in an entirely different context (Context-change group; Fig. 1C). We found that the Context-change group did not profit from the infantile spatial experience ($F(1, 27) = 7.490$, $P = 0.011$, for the difference between this group and the Spatial-experience group tested in the original context). In fact, the Context-change rats did not perform above chance level at any minute of the OPR retrieval phase at adulthood (all $t(11) > -0.759$ and $P > 0.160$, Fig. S4). These results corroborate the view that memories mainly containing contextual information of the infantile exposures helped the rats of the Spatial-experience group to enhance their OPR performance at adulthood testing.

### Adult rats show no episodic memory for spatial exposures during infancy

In a further control experiment, we asked whether the enhanced spatial memory capabilities in the Spatial-experience group were perhaps a direct consequence of an episodic memory that was formed for the individual exposures of the spatial experience during infancy and persisted into adulthood. To answer this question, we tested rats on an OPR task with the encoding phase performed during infancy (PD24), and the retrieval phase occurring only later, in adulthood (PD84, Long-term OPR group). Note, the exposure of the rat to only a single learning phase (on PD24, corresponding to the last day of exposures in the main experiment) represents the strictest test of the presence of detailed episodic-like memory for one solitary exposure. At the remote retrieval test in adulthood, the rats did not exhibit significant spatial memory (as measured by preferential exploration of the displaced object) at any minute of the retrieval phase (Fig. S5). In line with other studies[6,26,27], this result suggests that the rats did not retain a full-blown and detailed episodic representation—of the configuration of objects within its specific spatial context—over such a long time, thus excluding a persisting episodic memory as a cause for the Spatial-experience group's enhanced adult spatial performance.

### Infantile spatial experience forms memory in the prelimbic region of the mPFC

Based on studies of adults, the transformation of episodic memory during systems consolidation is thought to be mediated through a

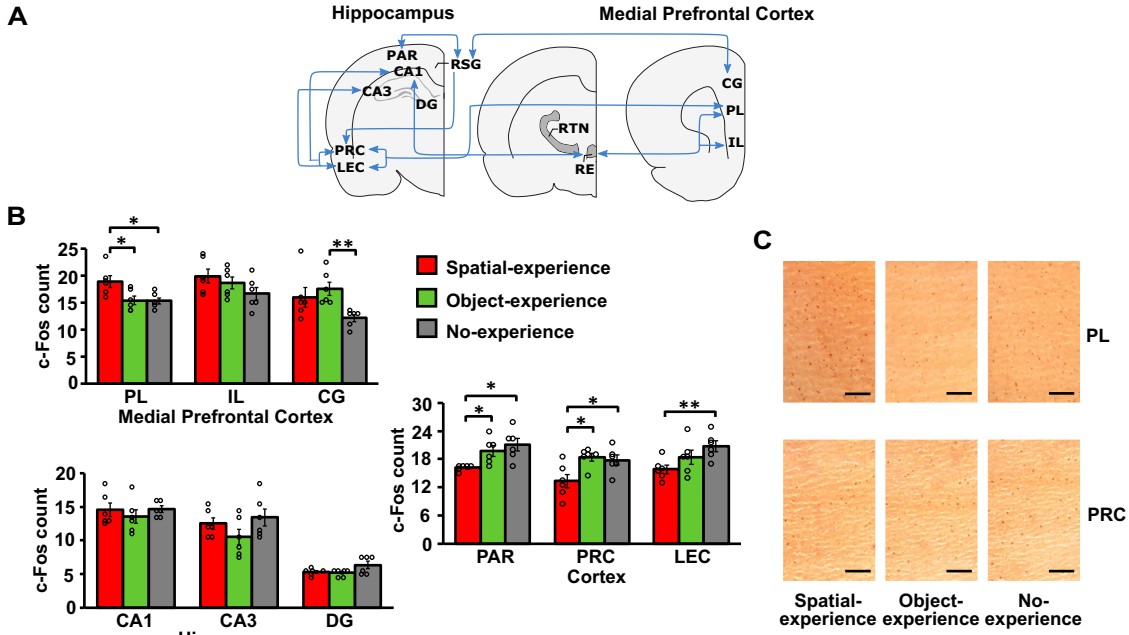

**Fig. 2 | c-Fos activity in cortical, thalamic and hippocampal regions of interest.**
**A** Schema of the selected regions of interest (ROI) contributing to memory formation in the hippocampus-dependent episodic memory system (modified from ref. 31). Medial prefrontal cortex: *PL* prelimbic, *IL* infralimbic, *CG* cingulate cortex; thalamus, *RE* nucleus reuniens, RTN reticular thalamic nucleus, posterior cortex: RSG retrosplenial, PAR parietal, PRC perirhinal, LEC lateral entorhinal cortices; hippocampus, CA1 cornu ammonis 1, CA3 cornu ammonis 3, DG dentate gyrus.

**B** Mean ± SEM counts of c-Fos+ cells in the Spatial-experience (red bars, $n = 6$), Object-experience (green bars, $n = 6$), and No-experience groups (grey bars, $n = 6$) in subregions of (top) the medial prefrontal cortex, (middle) posterior cortex and (bottom) hippocampus. *$P < 0.05$ and **$P < 0.01$ for pairwise comparisons (two-sided *t*-test) between experimental groups. **C** Representative images of c-Fos staining selected for cell count analysis in PL and PRC (scale bar: 150 μm). Source data are provided as a Source Data file.

dialogue between the hippocampus and mainly cortical networks, whereby the hippocampus initially binds the distributed representations of an experience into a coherent episodic memory representation[8,10,28–31]. Repeated reactivations of the neuronal representation support a gradual redistribution of the representation towards cortical networks eventually storing an abstracted version of the memory. It is in this context that we wondered to what extent the rats that experienced the spatial configuration change during infancy relied on neocortical and hippocampal networks when tested on the OPR task in adulthood. We examined expression of the activity-regulated gene c-Fos to map brain activity during adult OPR retrieval testing with a focus on hippocampal structures (CA1, CA3, dentate gyrus) and a thalamo-cortical system of regions well-known to contribute to the transformation of spatial episodic memory[31], and which essentially comprises the medial prefrontal cortex (mPFC, including prelimbic, infralimbic and cingulate cortices), the parietal cortex (PAR), the perirhinal cortex (PRC), and—as a structure essential to connecting hippocampal and cortical systems—the lateral entorhinal cortex (LEC) (Fig. 2A). Remarkably, this analysis did not show any difference in c-Fos activity between the Spatial-experience group and the control groups of Object-experience and No-experience in any of the hippocampal regions (all $F(2, 17) < 3.258$, $P > 0.067$, see Fig. 2B for pairwise comparisons). There were, however, major differences in cortical areas ($F(45.801, 274.806) = 5.518$, $P = 0.003$ for global ANOVA Group x Area interaction). The rats with spatial experience during infancy, after adult OPR retrieval, displayed enhanced c-Fos activity in mPFC, specifically in the prelimbic cortex (PL) ($F(2, 17) = 5.952$, $P = 0.013$). Concurrently, c-Fos activity was consistently decreased in the PAR, PRC, and also in the thalamic RE (Fig. S6). Note, all of these changes were significant in comparison with both the Object-experience and the No-experience group, as well as in comparison with a Home cage control group (Fig. S6) which remained in its home cage during OPR testing at adulthood. Moreover, the pattern of

increased c-Fos activity in the PL region of the mPFC but unchanged hippocampal activity characterizing the Spatial-experience group was also confirmed in a comparison with retrieval-related c-Fos activity in the Context-change group (Fig. S7).

Additional exploratory analyses of functional connectivity revealed that infantile experience, in general, reduced interregional c-Fos coactivation to a few distinct connections, in comparison with the No-experience control group (Fig. S8). The idea that stimulation during infancy non-specifically promotes the development of hippocampal memory systems[32] was also supported by analyses of cytochrome oxidase (COx) activity, a trait marker of neuronal activity reflecting the basal tissue energy demands, which was enhanced in the adult rats in hippocampus and neighbouring regions following both spatial and object experience during infancy (Fig. S9).

We wondered whether activity in the PL region is already enhanced at encoding of the spatial experience during infancy. To test this, we examined two additional groups of pups ($n = 6$ each, Fig. 3A), i.e., a Spatial-experience group that was subjected to the same procedures, with 4 exposures to changes in spatial configurations, as the Spatial-experience group of the main experiments, and a Home cage control group that remained in the home cage at the times of exposure to spatial experience during infancy. In both groups, c-Fos activity during the time of the fourth exposure (on PD24) was determined. As hypothesized, the pups of the Spatial-experience group displayed significantly enhanced c-Fos activity in the PL region of the mPFC, as well as in the neighbouring IL region. In addition, c-Fos activity was enhanced in the CA1 and DG region of the hippocampus ($F(12,120) = 2.750$, $P = 0.002$ for ANOVA Group x Area interaction, see Fig. 3B for pairwise comparisons). Overall, the observed engagement of the prelimbic mPFC at encoding of the spatial experience during infancy is consistent with the view that the representations used to enhance OPR performance of the Spatial-experience group at adulthood, were rather quickly formed already during infancy, although

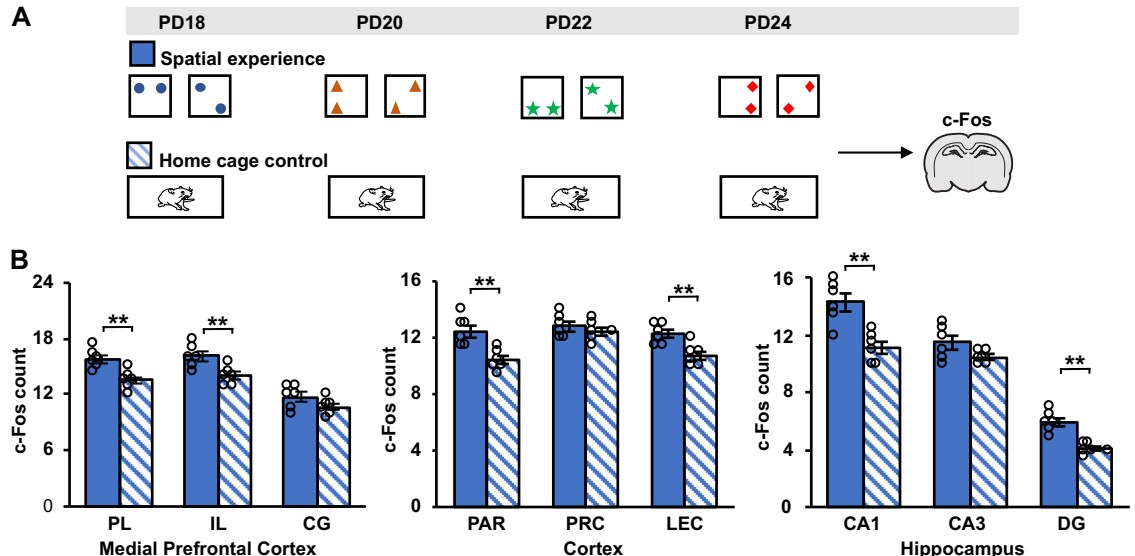

**Fig. 3 | Encoding of spatial experience during infancy is associated with activation of mPFC and hippocampus. A** Experimental procedures. The Spatial-experience group of this study (*n* = 6) was subjected to changes in spatial configuration on 4 days using the identical procedure as in the main experiments, and c-Fos activity was determined during the fourth exposure (on PD24). The Home cage control group (*n* = 6) remained in the home cage during the times of exposures. **B** Mean ± SEM c-Fos⁺ cell counts for the Spatial-experience (blue bars) and Home cage control groups (hatched bars), in (left) subregions of the medial prefrontal cortex, PL prelimbic cortex, IL infralimbic cortex, CG cingulate cortex, (middle), the parietal (PAR), perirhinal (PRC) and lateral entorhinal (LEC) cortices, and (right) in hippocampal subfields, CA1 cornu ammonis 1, CA3 cornu ammonis 3, DG dentate gyrus. **P* < 0.01 for pairwise comparisons (two-sided *t*-test) between groups. Source data are provided as a Source Data file.

these additional experiments in infant rats did not aim to specifically dissociate encoding of spatial experience (from object experience).

## Inhibiting the prelimbic region of mPFC at adult OPR testing abolishes benefits from early spatial experience

Our examination of c-Fos activity indicated increased activity at OPR testing during adulthood specifically in the PL region of the mPFC in the Spatial-experience group in comparison with both the Object-experience and No-experience control groups. The mere association of increased c-Fos with enhanced OPR performance, however, can be confounded by various factors (e.g., the differences in overt behaviour between the groups) and does not allow for valid conclusions as to the contribution of a specific brain region to the memory of interest. Therefore, to test whether representations in the PL region of the mPFC causally contribute to mediating the enhancing effects on spatial learning during adulthood, two further groups of rats were added which were subjected to the same protocol as the Spatial experience group of the main experiment, with four exposures to changes in spatial configurations. However, at retrieval testing of the OPR task at adulthood, in one of the groups (Muscimol) functioning of the PL mPFC was inhibited by the bilateral injection of the GABA receptor agonist muscimol, whereas the other group was injected with Saline (Fig. 4). Rats of the Saline group with a fully functioning PL region of mPFC displayed robust OPR memory (*t*(17) = 3,831, *P* < 0.005, one-sample *t*-test against chance level for first 3 min of test) replicating findings of the main experiment. By contrast, OPR performance of the Muscimol group with an inhibited PL region remained below chance level (*t*(17) = −0.129, *P* = 0.698). The impairing effect of PL inhibition was also significant in comparison with the Saline control group (*F*(1, 34) = 8.4, *P* < 0.01, Fig. 4). Total exploration time and total distance travelled at retrieval testing were comparable between groups (*F*(1,34) = 0.76 for distance travelled, *F*(1,34) = 2.87 for total exploration time, *P* > 0.1, one-way ANOVA).

Although these results support the view of the PL region of the mPFC being critically involved in mediating the improving effects of infantile spatial experience on OPR performance in adulthood, it could

be argued that, in these experiments, the suppression of the PL mPFC by muscimol basically and independently of any prior infantile experience interfered with OPR performance. To exclude this possibility, we performed additional control experiments in a new group of naïve adult rats (-PD80) using the same procedures as in the main experiments except that the OPR retention interval between encoding and retrieval testing was shortened to 18 min (which inevitably lowered memory requirements but represent conditions where adult rats normally show significant OPR memory). In these experiments, the rats showed significant OPR memory after infusion of vehicle as well as after muscimol (*t*(23) > 2.456, *P* < 0.022, for both conditions, with no difference between conditions, *P* = 0.41, Fig. S11).

In combination, results from these experiments indicate that the PL mPFC plays a causal role in mediating the improving effects of infantile spatial experiences on the adult rat's spatial learning capabilities. In conjunction with the selectively enhanced c-Fos activity in the PL region but unchanged c-Fos activity in hippocampal regions at OPR testing during adulthood in rats with infantile spatial experience, our findings, indeed, support the view that the exposure to spatial experience forms contextual representations that reside in mPFC rather than hippocampal networks, and that are used to improve spatial learning in similar contexts during adulthood.

## Effects of early spatial experience on adult OPR performance depend on developmental age

Is infancy a period when the brain is particularly capable of forming contextual spatial long-term memory? The first years of life are characterized by distinct conditions of synaptic plasticity and the shaping of neuronal circuits mediating memory formation, and many of these conditions, like a strongly elevated neurogenesis, particularly apply to the hippocampus, i.e., the structure centrally involved in the formation of spatial representations[14,32–34]. In light of these pervasive alterations in hippocampal memory function during infancy, we tested the effects of early spatial experience on adult OPR performance at two further ages, i.e., in addition to a group of rats exposed to the spatial configuration change during Infancy (PD18-PD24), in two other groups this

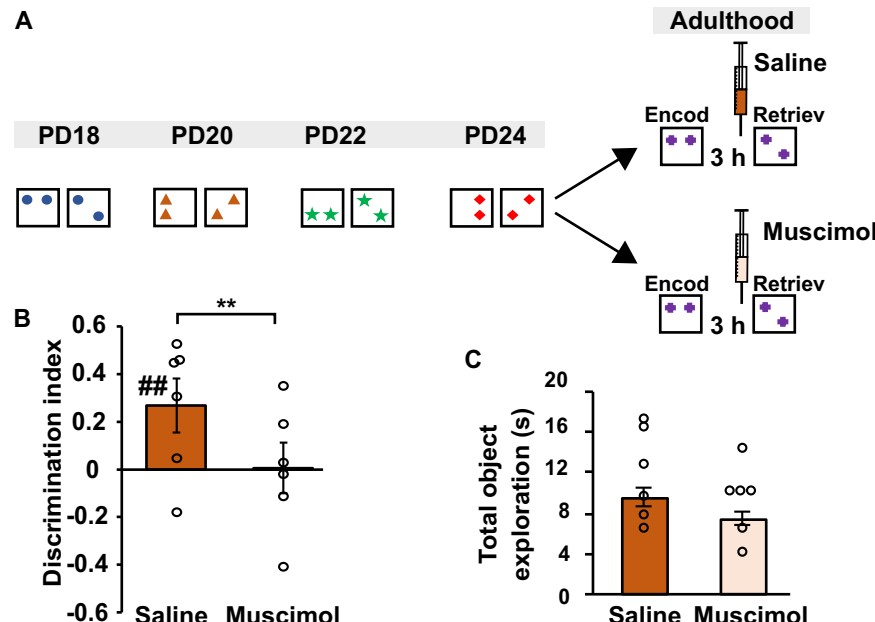

**Fig. 4 | Inhibiting the PL region of mPFC at adult OPR testing abolishes benefits from early spatial experience. A** Experimental procedures: During infancy, pups were subjected to changes in the spatial configuration on 4 days using the identical procedure as in the Spatial-experience group of the main experiments. At adulthood, rats received bilateral cannula implantation on the PL region of the mPFC and were randomly assigned to a Saline or Muscimol group. After recovery (>14 days), both groups were tested on the OPR task with, muscimol or saline injected 18 min before the retrieval phase. **B** Mean ± SEM values for OPR memory, i.e., discrimination ratios (n = 6) and **C** Total object exploration time at adulthood testing. Mean ± SEM values for the first 3 min of retrieval phase, dot plots overlaid (n = 6). $^{\#\#}P = 0.001$, one-sample t-test (two-sided) against chance level; $^{**}P = 0.007$ for difference between groups (one-way ANOVA). Note, significant OPR memory in Saline group is abolished after muscimol. (Fig. S10, shows discrimination ratios for the entire 5 min retrieval phase and further control parameters). Source data are provided as a Source Data file.

exposure took place during periods corresponding to Early childhood (PD25-PD31) and Adolescence (PD48-PD54), respectively. Note, these experiments comprised new groups of rats for the No-experience condition and the Spatial-experience condition which replicated the effects of the main experiment OPR performance at adulthood testing was indistinguishable between the Early childhood and Adolescence groups ($t(26) = -0.053, -0.112, 0.033, 0.231$ and $-0.074$, all $P > 0.819$ for pairwise comparison per minute), but was significantly worse than that of the infantile Spatial-experience group with this group difference being strongest for the 1st min of the retrieval phase ($F(1.883, 212) = 5.413$, $P = 0.007$ and $P = 0.36$, for Group x Minutes interaction and Group main effect, respectively, in an ANOVA with pooled Early childhood and Adolescence groups, Fig. 5A). Thus, the spatial experience benefitted adult OPR memory only in the pups exposed to this experience during infancy, whereas spatial experience occurring later during early childhood or adolescence left adult OPR performance entirely unchanged when compared with the No-experience control group ($P = 0.015$, for the planned contrast of the Infancy group with all other groups; Fig. 5A). Overall, the pattern of results agrees with our hypothesis that infancy is a period of particular sensitivity to spatial experiences and for taking them to build contextual long-term memories[7]. However, in light of the relatively small group sizes these conclusions remain tentative.

## Effects of early spatial experience on adult OPR memory depend on post-experience sleep

Sleep supports memory consolidation[29,35,36]. Sleep-dependent consolidation is thought of as an active systems consolidation process that critically depends on hippocampal function[29], and in which the repeated neuronal replay of newly encoded memories promotes the gradual transformation of memories into persistent and more abstract long-term memories[29,36,37]. We therefore asked whether the benefit from spatial experience during infancy for adult spatial learning

capabilities depends on the occurrence of sleep after the infant experience. We compared the Spatial-experience group of rats which all showed normal extensive sleep after each exposure to the arena during infancy, with a Sleep-deprivation group of rats which were subjected to a 90 min awake period following each of these experiences. At adult OPR testing, performance of the Sleep-deprived rats was significantly worse than that of the Spatial-experience group ($t(38) = 3.042, P = 0.004$ for the first minute; $F(1, 38) = 4.933, P = 0.032$ for Group × Minute interaction), and did not differ from that of a No-experience control group ($t(23) = 0.831, -0.282, -0.783, -1.424,$ and $-1.105$, all $P > 0.289$, for all comparisons; Fig. 5B). In parallel, preventing sleep after the infantile spatial experience nullified the enhancement of retrieval-related c-Fos activity in the prelimbic area of the mPFC characterizing the Spatial-experience group ($t(10)) = 2.768, P = 0.020$, for pairwise comparison between groups.

It is unlikely that these effects of sleep deprivation were confounded by stress capable to induce general adverse effects on brain development, as sleep deprivation of the pups was performed very cautiously using gentle handling in the presence of the littermates and close to the mother that could be seen and smelt by the pup. Moreover, duration of sleep deprivation was relatively short (1.5 h), and none of the pups showed any behavioural signs of stress or fear during sleep deprivation. The use of similar deprivation procedures in adult rats remains without any effect on stress hormone levels like corticosterone (e.g., ref. 38). Thus, assuming that the sleep deprivation procedure did not induce substantial side effects, our findings support the conclusion that the beneficial effect of infant spatial experience on adult spatial learning requires sleep to occur after the infant experience.

## Discussion

We present a novel approach that seeks to characterize the influence of discrete non-emotional spatial experience during infancy on spatial

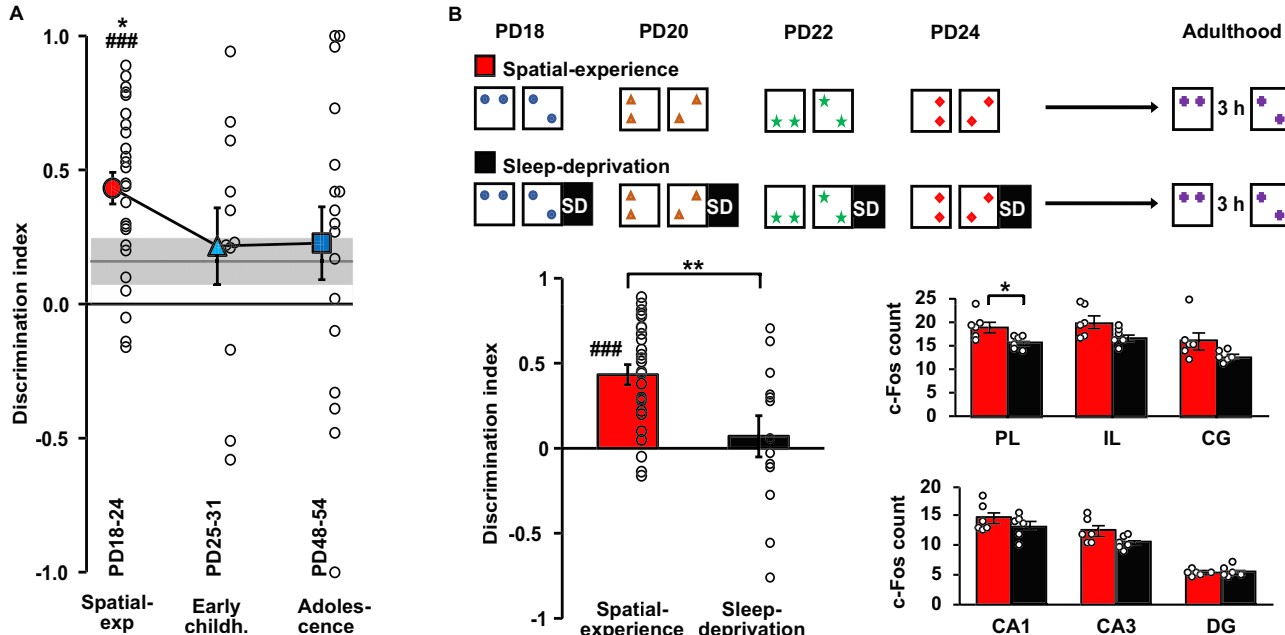

**Fig. 5 | Developmental trajectory of the effects of spatial experience and the role of post-experience sleep. A** OPR memory (mean ± SEM discrimination ratios during 1st min of retrieval phase, dot plots overlaid) at adulthood testing for the Spatial-experience group exposed to spatial experience during infancy (PD18-PD24, $n = 27$), for the Early childhood group ($n = 11$) and the Adolescence group ($n = 17$). OPR memory for the No-experience control group is indicated by horizontal line (± SEM grey shaded area). *$P = 0.015$ for planned contrast with all other groups, ### $P = 0.000$ for one-sampled $t$-test (two-sided) against chance level, $F_{(1, 56)} = 1.318$, $P = 0.256$, for comparisons of Early childhood and Adolescence groups with the No-experience control. Note only the pups exposed to spatial experience between PD18-24 exhibit significant OPR memory that, in addition, is significantly stronger in comparison with all other groups. **B** Top−Procedure: Pups either had undisturbed sleep ($n = 27$) or were deprived from sleep (SD) ($n = 13$) in the 90 min interval (black) after spatial experience. Bottom left−mean ± SEM OPR memory for the two groups (dot plots overlaid), **$P = 0.004$ for differences between groups ($t$-test, two-sided) and ###$P = 0.000$ for one-sampled $t$-test (two-sided) against chance level. Bottom right−counts of c-Fos+ cells in subgroups normal sleeping ($n = 6$ rats) and sleep deprived ($n = 6$ rats) animals in (top) subregions of the medial prefrontal cortex, PL prelimbic cortex, IL infralimbic cortex, and CG cingulate cortex, and (bottom) in hippocampal subfields, CA1 cornu ammonis 1, CA3 cornu ammonis 3, and DG dentate gyrus. *$P = 0.020$ for pairwise comparisons (two-sided $t$-test) between experimental groups. Sleep deprivation nullified the enhancing effect of spatial experience during infancy on OPR performance during adulthood, along with the retrieval associated enhancement in c-Fos activity in the prelimbic PL region of the mPFC. (see Methods). Source data are provided as a Source Data file.

learning in adulthood. We find that a seemingly insignificant event, i.e., a change in the spatial configuration of two objects the rat is exposed to in an experimental arena during its infancy for a few times − overall no more than 20 min and in the absence of any rewarding or aversive stimulation − distinctly impacts learning behaviour and related brain organization during adulthood: This means at adulthood, the rats displayed enhanced capabilities to form spatial memories. Notably, the adult rats' memory capabilities were similarly enhanced when during infancy the rats were exposed to the same two objects presented also twice on each exposure but, in the absence of a configurational change, suggesting that the memories producing the adult rats' enhanced spatial memory capability were based on contextual information rather than the experience of the change in the object configuration. The enhancement in spatial memory performance in adulthood was specifically related to the use of representational systems residing in the prelimbic region of the medial prefrontal cortex (mPFC) rather than hippocampal networks. Our findings, moreover, suggest that such discrete spatial events have the power to form memory for adulthood, only when experienced during infancy, but not during childhood or adolescence, and only when the experience during infancy is followed by sleep, underlining the importance of infant sleep for forming lifelong memory[35,39].

Our study identified the prelimbic mPFC as a region that is critical for mediating the enhancing effect of spatial experience during infancy on the adult rat's spatial learning capability. Studies of adult rodents have indeed revealed the mPFC and, particularly, its prelimbic subregion as a hub area mediating remote episodic memory recall[40-45]. In the mPFC including prelimbic cortex spatial representations rapidly

emerge even in conditions of free exploration[46,47] which is consistent with our findings in the rat pups showing increased c-Fos levels in prelimbic and neighbouring mPFC right after the last spatial experience during infancy. Thus, engaging already at the time of spatial encoding, mPFC areas are increasingly activated with increasing age of the memory while hippocampal activity remains unchanged[40,48,49]. For remote recall, intracortical projections from this region may activate representations that, in the spatial domain, involve specific circuitry in the parietal and perirhinal cortices[31,50]. That the c-Fos response to OPR retrieval testing at adulthood in these more posterior cortices was partly lower in our Spatial-experience group than in the two control groups is difficult to integrate but, might reflect a contraction and sharpening of the representations that can occur as a consequence of long-term experience[51]. As to the hippocampus, the lack of differences in c-Fos expression in these networks at adult OPR testing well agrees with previous studies showing that the hippocampus is crucial for forming spatial memory, yet does not store long-lasting memories[52-54].

What is the content of the memory formed during infantile spatial experience, that enhances memory capabilities at adulthood? To address this core question of our behavioural experiments we examined adult rats' OPR performance in two additional conditions, i.e., the Stationary-experience and the Context-change conditions. The infantile exposures in the Stationary-experience group lacked the experience of a change in the spatial configuration of the two objects but were otherwise identical to those of the Spatial-experience group. Importantly, the exposures took place in the very same environmental arena context as that used for the pups of the Spatial-experience group. Thus, the rats of the Stationary-experience group benefitting

(at OPR testing in adulthood) from the infantile exposures to the same degree as the Spatial-experience group strongly suggests that the enhanced OPR performance these rats showed in adulthood, resulted from memories formed of the arena context, whereas the mere experience of a change in the configuration between the objects seems to be of secondary relevance. This view that rats of the Spatial-experience group relied on memory of contextual information about the infantile exposures at adult OPR testing, is corroborated by our findings in the Context-change group. These animals did not show an enhanced OPR performance at adult testing although during infancy they had been exposed to the same spatial experience as the Spatial-experience group. Yet, OPR testing at adulthood took place in a context entirely different from that experienced during the exposures at infancy, i.e., in a different room with different distal cues, a different floor and with a different experimenter. Accordingly, we assume that the rats of the Spatial-experience group benefited from an enduring memory of such spatial and social features of the environmental context that belonged to the infantile exposures and acted as reminders when they were tested on the OPR task as adults.

Although our behavioural findings conclusively support the view that the improved adult OPR performance of the Spatial-experience group was based on contextual memories of the infantile exposures, this explanation is difficult to reconcile with the findings in the Object-experience group: This group did not benefit from the infantile exposures at adult OPR testing, although their exposures during infancy took place in the same environmental context as in the Spatial-experience group. Instead of a change in the spatial configuration of two identical objects, rats of the Object-experience group were exposed to a change in one of the two objects, i.e., at the second arena visit of each exposure one of the objects was replaced by a novel object. Although we did not find behavioural signs of an increased interest in the objects during the infantile exposures in these pups (Fig. S3), one explanation for their lack of benefit at adult OPR testing could be that during their infantile exposures the pups were distracted by the novel objects from encoding the relevant context information. Indeed, the conditions are reminiscent to those found in 15–20 months old human infants who did not discriminate the spatial room context while searching for toys hidden in boxes but, surprisingly, showed the ability to disambiguate the boxes according to the room context when the toy cues were absent[55]. Nevertheless, the interpretation that the novel objects distracted the rat pups of our Object-experience group from encoding spatial context remains tentative and needs to be scrutinized in further experiments.

Our finding indicating that benefits for spatial capabilities in adulthood originate from memory of contextual information formed during infantile experiences well fits the observation that the memory recall in these early years of life shows a distinctly greater context-dependency than in adolescence and adulthood[56]. Of note, rat pups have been shown to encode contextual cues into mPFC regions including the prelimbic region, from early on (i.e., on PD16)[57]. Here, c-Fos activity in the same mPFC region was enhanced above levels in home cage controls already after encoding of the fourth spatial exposure, suggesting that contextual information forms part of the supraordinate representation mediating the enhancing effects of infantile spatial experience on adult learning capabilities. A preferential formation of persisting contextual memories in mPFC networks might be advantageous as such memories might serve as reference frames, not only scaffolding the recall of multiple episodes experienced in the same or similar context (e.g., refs. 58–60) but also effectively guiding future behaviour in such context, thereby supporting—in a context-dependent manner—learning processes like those seen in our Spatial-experience group.

While in the mature brain, the scaffolding of detailed memory recall by spatial context information has been commonly linked to hippocampal function (e.g., ref. 61), the hippocampus is assumed to be developing and not fully functioning during infancy. In fact, both the strong contextual dependency of memory recall as well as the tendency to form more generalized memories during infancy, have been ascribed to the hippocampus being still immature and, e.g., less able to differentiate context and event[11,14,33,35,39]. Against this backdrop, a plausible reason for the preferential encoding and storage of context information in infants could be its greater invariance across episodes, in comparison with the individual events. Thus, the infantile experience for our Spatial-experience group comprised four different changes in the spatial configuration presented with different pairs of objects at each of the four exposures, while the arena context remained the same across all four exposures. Whatever the case, even more important is that the contextual representations mediating the benefit in spatial memory performance in adulthood were apparently formed in mPFC rather than hippocampal networks. Beyond our demonstration of a causal contribution of the prelimbic mPFC to the enhancement in adult spatial learning capabilities, the rats with infantile spatial experience displayed enhanced c-Fos activity in prelimbic and neighbouring mPFC regions, but not in hippocampal areas at adult OPR testing. In light of these findings, it is tempting to speculate that the contextual representations in the mPFC that are presumably formed rapidly upon infantile spatial experience also serve to advance, in a top-down manner, maturation of hippocampal networks such that these networks become gradually shaped towards the preferential processing of context congruent experience[7,62].

Our findings underline the importance of sleep following the exposures to spatial experience for forming such long-lasting representations in cortical networks during a sensitive period very early in life, and it may well be the immature conditions in the hippocampus that favour the impact of post-encoding sleep during this sensitive period towards the fast formation of a supraordinate contextual memory in cortical networks[12,39]. Sleep has been shown to be critical for the quick transformation of experience encoded in the hippocampus-dependent episodic memory system, into less detailed representations mainly residing in cortical networks[63], and such transformation might comprise the simultaneous forgetting of the episodic, presumably hippocampal, representation. In this view, sleep might also contribute to the forgetting of episodes experienced during infancy, as it was observed in the rats of our Long-term OPR group (Fig. S5) and as it constitutes the phenomenon of infantile amnesia[35,64,65]. However, rather than infantile amnesia and the forgetting of episodic memory during infancy, the central question addressed by our study was about the information experienced in early-life that is *not* forgotten but maintained in memory for the long term. In this regard, our findings suggest that, mediated by a sleep-dependent mechanism, the infant brain preferentially forms long-term memories for contextual information which critically involve mPFC networks.

## Methods
### Animals and experimental groups
A total of 191 male Long-Evans rats were used for the experiments. Rat pups were allocated to the different experimental groups such that each group derived from 1–3 litters of 3–6 pups. In total 35 litters were used for the whole study. Two litters were born in our own animal facilities (each litter was culled to 4 pups one or 2 days after parturition). The remaining pups arrived (from Janvier, Le Genest-Saint-Isle, France) in our facilities at least 4 days before any manipulation in order to allow acclimatization. It was ensured by inspection that all pups had opened their eyes and already started to explore their home cage surroundings on the day the experimental procedures started. The pups were maintained with their dam until weaning (PD21). The animal colony was kept at a room temperature of $22 \pm 1\,°C$, on a 12 h/12 h light/dark cycle (lights on at 6:00 h). All rats had free access to food and water throughout the experiments. All experimental procedures were performed in accordance with the European animal protection laws

(Directive 2010/63/EU, European Community) and were approved by the Baden-Württemberg state authority.

The 3 groups of the main experiments were the Spatial-experience group ($n = 18$ pups), the Object-experience group ($n = 18$ pups), and the No-experience group ($n = 18$ pups). To dissociate the effects of configurational changes and context in the infantile experiences, an additional group (Stationary-experience, $n = 12$ pups) was tested in the same conditions as the Spatial-experience group, except that both objects remained at the same location across each pair of visits during infancy. A further control group included rats tested at adulthood in a different context (Context-change group, $n = 12$ pups). To test for long-term episodic memory for early spatial experience, another separate group (Long-term OPR, $n = 12$ pups) was exposed to a single encoding phase on the OPR at PD24 and tested at adulthood. Exposure to spatial and object-experience took place between PD18 and PD24 with, this interval selected based on evidence that the pups' capabilities for object exploration are not firmly established before PD18[25] and that maturation of the hippocampus proceeds at least until PD24[5,66]. c-Fos activity during OPR retrieval testing at adulthood was compared to that of a Home cage control group of rats ($n = 6$). In two further groups of pups (each $n = 6$), c-Fos activity was assessed during infancy on PD24, in one group right after the last of the four exposures to spatial experience, with the other remaining in the home cage during the four arena visits.

Two additional groups of rats (each $n = 7$) were used to investigate the role of the prelimbic region of the mPFC during adult OPR retrieval after infantile spatial experience. Both groups were exposed to the same spatial experiences during infancy as the Spatial-experience group of the main experiment. Prior to OPR retrieval testing at adulthood, rats of one of the groups were injected with muscimol to suppress functioning of this area, whereas the other group received saline. A third group of rats ($n = 4$) was used here to exclude that suppressing the prelimbic mPFC basically interferes with OPR performance. Each of these rats were tested twice on the muscimol and saline conditions using the same procedures as for the two other groups, except that these rats were not exposed to any prior infantile experience and that the interval between encoding and retrieval was shortened to 18 min.

In two separate experiments, the developmental trajectory of the effects of early spatial experience on adult OPR performance and the role of sleep after infantile spatial experience were examined. For these experiments, two groups of pups were exposed to the early spatial experience manipulation during Early childhood (starting from PD25, $n = 11$ pups) and Adolescence (from PD48, $n = 17$ pups), respectively. A Sleep-deprivation group of pups ($n = 13$) was deprived of sleep after exposure to the early spatial experience during infancy. For these two separate experiments, also the Spatial-experience and No-experience control groups were newly formed to avoid multiple testing against the same reference groups. Furthermore, we aimed at enhancing statistical power as based on forgoing studies[25], we expected increased response variability with the inclusion of the Early childhood and Adolescence groups. Two new sets of rats (each, $n = 12$) were subjected to the same procedures as described for the Spatial-experience and No-experience groups of the main experiment, and it was assured (i) that the target effect of an enhanced OPR memory was replicated in these animals ($F(1, 20) = 5.437$, $P = 0.030$, for the difference in OPR memory between groups) and (ii) that the groups in OPR memory performance did not differ from the respective groups of the main experiments ($P > 0.642$, for all independent $t$-test comparisons on each minute of the OPR memory test). For statistical comparison the new sets were combined with the respective groups of the main experiments.

### Behavioural procedures

All procedures were performed between 7:00–16:00 h (i.e., the light phase). All animals exposed to early experience manipulations

received 5 sessions of 5 min handling on 5 (or 3, in 73 cases) consecutive days. For these groups (except the animals exposed to the early spatial experience at adolescence, i.e., PD48) the handling procedures included the dam in order to diminish potential stress. On each of the following 3 days, a 10 min habituation session was performed where the rats were allowed to freely explore the empty arena. In each session, they were introduced into the arena from a different side to support allocentric mapping. After the session, the rats were returned to the dam in the home cage.

On the day after the last habituation session, the rats of the Spatial-experience and Object-experience groups were subjected to the early spatial and object experience, respectively. For the Spatial-experience group (and related control groups) this experience consisted of two 5 min visits of the arena in which two identical objects were placed. At the second of these visits one of the objects was moved to another place (Fig. 1A). The 2 visits were separated by a 5 min break (for which the pup was brought to the home cage and dam) and, were repeated (with different objects and changes in spatial configuration) on 3 succeeding days (every other day). For the Object-experience group, instead of displacing one of the objects, one of the objects was replaced by another during the second of the two visits. At each visit, the rat was introduced into the arena from a different side with the rat always facing the respective wall of the arena. Behavioural responses to the change in the spatial vs. object configuration were comparable in terms of the time the animals spent exploring both objects ($F(1, 33) = 1.549$, $P = 0.222$, across the 4 days) and travelling in the arena ($F(1, 33) = 1.515$, $P = 0.227$). The No-experience control animals did not receive any experimental manipulation (including handling, exposures to the arena) during infancy.

At adulthood, all groups were tested on an Object-place recognition (OPR) task. This task which had been employed in an identical manner in a forgoing study[25] was designed such that recognition performance under baseline conditions was expected to be low. Specifically, to make the task more challenging, a long 3 h delay between encoding and test was used in combination with a rather short 5 min encoding period. Encoding difficulty was further enhanced by the use of an open field with only a few cues inside the arena enforcing the animal to navigate based on distal cues outside the arena (see section Apparatus and objects). During the 3 h retention interval the rats were moved to the home cage. Preparation for OPR testing included exactly the same procedures of 5 min handling (on 3 days), followed by 10 min sessions to habituate the animal to the empty arena (on 3 days). For the Long-term OPR control group, the encoding phase took place during infancy (PD24) and recognition was tested at adulthood (PD84), with handling and habituation procedures only preceding the infantile encoding phase. In the interval between the "early spatial experience" manipulation and adult OPR testing, the animals were kept in their home cages and weighted weekly, but otherwise remained without any experimental stimulation.

### Apparatus and objects

For establishing the early "spatial" and non-spatial "object" experience manipulation a quadratic dark grey open field was used (43 cm × 43 cm, height of walls 35 cm). Objects (height 10–18 cm) were glass bottles of different shapes, filled with water or sand of different colours. They had sufficient weight to ensure the rats could not displace them. For each of the four exposures different objects were used placed at different locations, with the order of objects used and their location in the arena at a specific exposure counterbalanced across the rats of the group. Also, the same objects were used for the Spatial-experience and Object-experience groups in a counterbalanced way. To support allocentric spatial mapping, a number of distal cues were available: The North side of the arena was headed towards a white wall whereas the East and West sides were surrounded by a grey curtain. The South side of the arena faced a removable black curtain (which also served as

the experimenter's entrance). Additional discrete distal cues were provided on the ceiling: a brown wood square (40 cm × 40 cm) located 120 cm above the open field and 36 cm below the ceiling. At two sides, a pink ball (10 cm diameter) and a light-brown cartoon box (25 cm × 25 cm × 10 cm), respectively, were attached to the curtains. Two fluorescent strip lights placed on the floor of the room provided indirect light. White noise was presented at a constant intensity during all procedures, to mask any disturbing sounds.

OPR testing during adulthood took place in a similar but larger arena (77 cm × 77 cm, height of walls 37 cm), compared with that during early spatial experience. Objects were also larger (22–29 cm) and quite different from those during early experience. Distal cues were as described for the early experience, except for the Context-change group. The context of OPR testing for this group differed in several aspects: The experimenter was a different person (all experiments were conducted by women), the experimental room was different and all distal cues as well as texture (wrinkled vs smooth floor) differed. Objects and arena were cleaned thoroughly between trials with 70% ethanol solution.

## Data collection and analysis
The rat's behaviour was video-recorded during the visits of the early experience as well as during the encoding and retrieval phases of the OPR task and visually scored offline by an experienced experimenter using the ANY-Maze tracking software (Stoelting Europe, Dublin, Ireland). Scoring was performed in a blinded manner with the scorer being unaware of which object was the familiar and the displaced object. Exploration was defined by the rat directing its nose to the object and sniffing. Climbing on an object or sitting next to it without any signs of active exploration was not included.

On the OPR task, allocentric spatial memory was analysed using the object discrimination index which is the standard way to assess OPR memory in adult rats, and is defined by the formula: DI = [( exploration time for novel object-location − exploration time for familiar object-location) / (exploration time for novel object-location + exploration time for familiar object-location)]. Additionally, the total time of object exploration (across both objects) and the total distance travelled during encoding and retrieval phases were determined.

## Sleep and sleep deprivation
The pups of the Spatial-experience group (as well as the pups of the Object-experience group) were returned to their home cage with their littermates and dam immediately after each second visit to the arena on PD18, PD20, PD22 and PD24. Assessment of (video-recorded) behaviour assured that the pups spent a minimum of 44 min of the 90 min post-experience interval in a sleeping position, close to the dam and often fully covered by the dam's body. Visual inspections performed in a separate sample of 6 pups confirmed the presence of sleep (i.e., closed eyes and occasional suckling) during the times the pup was mostly covered by the dam's body. The rats of the Sleep-deprivation group were deprived from sleep during the 90 min post-experience interval applying a "gentle-handling" procedure. The procedure was initiated as soon as the litter huddled together or one of the pups showed signs of sleep (e.g., taking a sleep posture or closing its eyes), and consisted of tapping on the cage, gently shaking the cage, disturbing nest-building behaviour and, to avoid huddling, separating the pups by placing them away from their littermates. During the post-experience intervals on PD18 and PD20 (preweaning), the dam was kept in a neighbouring cage allowing that the pups to see and smell her.

## c-Fos immunocytochemistry
After completing the retrieval phase of the OPR task, the rats were returned to their home cages for 90 min. Then, the animals were decapitated, and the brains were removed intact, frozen rapidly in methylbutane (Sigma-Aldrich, Taufkirchen, Germany), and stored at −40 °C (decapitation was used to also enable determination of COx). A 90 min delay after retrieval testing was used because c-Fos protein activity peaks with a latency of ~90 min after the event of interest[61]. The brains were coronally (30 μm) sectioned at −20 °C in a cryostat microtome (model HM 505-E, Microm International GmbH, Heidelberg, Germany). The sections were mounted on gelatinized slides for c-Fos immunocytochemical analysis, and on non-gelatinized slides for additional cytochrome oxidase (COx) histochemistry (see below). We defined the regions of interest (ROIs) based on the literature about cortical, thalamic and hippocampal regions involved in the formation of spatial and episodic memory formation[31], and determined the anatomically according to Paxinos and Watson's atlas[67]. ROIs and their distance (in mm) from bregma were: +3.24 mm for the prelimbic (PL), infralimbic (IL), and cingulate cortices (CG); −3.96 mm for the agranular (RSA) and granular retrosplenial cortex (RSG), for the parietal (PAR), perirhinal (PRC) and lateral entorhinal cortices (LEC); −1.80 mm for the thalamic nucleus reuniens (RE) and reticular thalamic nucleus (RTN); −3.96 mm for the hippocampal cornu ammonis 1 (CA1), cornu ammonis 3 (CA3) and dentate gyrus (DG) subfields.

## Brain processing
For c-Fos immunocytochemistry, the sections were post-fixed in buffered 4% paraformaldehyde (0.1 M, pH 7.4) for 30 min and rinsed in phosphate-buffered saline (PBS, 0.01 M, pH 7.4). They were subsequently incubated for 15 min with 3% hydrogen peroxidase in PBS to remove endogenous peroxidase activity, and then washed twice in PBS. After blocking with PBS solution containing 10% Triton X-100 (PBS-T, Sigma, USA) and 3% bovine serum albumin for 30 min, sections were incubated with a rabbit polyclonal anti-c-Fos solution (1:10,000, # sc-52, Santa Cruz Biotech, Santa Cruz, CA, USA) diluted in PBS-T for 24 h at 4 °C in a humid chamber. Slides were then washed 3 times with PBS and incubated in a goat anti-rabbit biotinylated IgG secondary antibody (diluted 1:200 in incubating solution, #31820, Thermo Scientific Pierce, Rockford, IL, USA) for 2 h at room temperature. They were washed for another 3 times in PBS and then reacted with the avidin biotin peroxidase complex (Vectastain ABC Ultrasensitive Elite Kit, Pierce, USA) for 1 h. After 2 more washes in PBS, the reaction was visualized by treating the sections for 3 min in a nickel-cobalt intensified diamino benzidine kit (Pierce, USA). The reaction was terminated by washing the sections twice in PBS. Slides were then dehydrated through a series of graded alcohols, cleared with xylene, and cover-slipped with Entellan (Merck, USA) for microscopic evaluation. All immunocytochemistry procedures included sections that served as controls where the primary antibody was not added. Slides containing sections of a specific brain region were stained at the same time. The experimenter performing the c-Fos and COx analyses was blind to the experimental conditions of the individual brains.

The total number of c-Fos positive nuclei was quantified in two alternate sections 30 μm apart. Quantification was performed by systematically sampling each of the regions selected using superimposed counting frames. Sizes of the counting frames ranged from 40,000 μm$^2$ (RE) to 360,000 μm$^2$ (PAR). The total area covered by these frames per region in each section was: 40,000 μm$^2$ in RE; 90,000 μm$^2$ in DG; 120,000 μm$^2$ in RTN and CA1; 180,000 μm$^2$ in RSA, RSG and CA3; 270,000 μm$^2$ in IL, PL, CG, PRC, and LEC; and 360,000 μm$^2$ in PAR. Cell counts were conducted using a microscope (Leica DFC490, Germany) coupled to a computer with software installed (Leica application suite, Germany). c-Fos positive nuclei were defined based on homogenous grey-black stained elements with a well-defined border. Finally, the mean count of two adjacent sections was calculated for each subject/brain and region.

## Cytochrome oxidase (COx) histochemistry

As an estimate of basal metabolic rates in the ROIs, we additionally measured COx activity using quantitative histochemistry[68,69]. To quantify enzymatic activity and control staining variability across different baths, sets of tissue homogenate standards from rat brains were cut at different thicknesses (10, 30, 50 and 70 µm), and included in each COx staining bath together with the experimental brain sections. The batch standards of brain homogenate were previously analysed by spectrophotometrical methods to measure mean COx activity and were used to generate a single regression equation between COx activity and the optical density of the experimental sections, as reference for the comparison of the different tissues (see below). The sections and standards were incubated for 5 min in 0.1 phosphate buffer with 10% (w/v) sucrose and 0.5 (v/v) glutaraldehyde, pH 7.6. Next, they were immersed in 3 batches of 0.1 M phosphate buffer with 10% (w/v) sucrose were given for 5 min each. Subsequently, 0.05 M Tris buffer, pH 7.6, with 275 mg/l cobalt chloride, 10% (w/v) sucrose, and 0.5 (v/v) dimethyl-sulfoxide was applied for 10 min. Then, sections and standards were incubated in a solution of 0.06 g cytochrome c, 0.016 g catalase, 40 g sucrose, 2 ml dimethyl-sulfoxide, and 0.4 g diamino-benzidine tetra-hydrochloride (Sigma-Aldrich, Madrid, Spain) in 800 ml of 0.1 M phosphate buffer at 37 °C for 1 h. The reaction was stopped by fixing the tissue in buffered formalin for 30 min at room temperature with 10% (w/v) sucrose and 4% (v/v) formalin. Finally, the slides were dehydrated, cleared with xylene, and cover-slipped with Entellan (Merck, Darmstadt, Germany).

The COx histochemical staining intensity was quantified by means of densitometric analysis, using a computer-assisted image analysis workstation (MCID, Interfocus ImagingLtd., Linton, England). The mean optical density (OD) of each region was measured in three alternate sections, 30 µm apart. In each section, four non-overlapping readings were taken, using a square-shaped sampling window adjusted for each region size. A total of twelve measurements were taken per region. Calibration of OD measures for COx activity units was performed using the stained homogenate standards for each staining batch. For each staining batch the software calculated a linear regression between optical density and COx activity, using the measured OD attributed to each section. Average relative OD measured in each brain region was converted into COx activity units (1 unit: 1 µmol of cytochrome c oxidized/min/g tissue wet weight at 23 °C) using the calculated regression curve in each homogenate standard. The linear regression equations calculated to estimate COx activity from OD measures in the brain sections were also used to assess inter-batch variability which was <1%.

## Inactivating the prelimbic region of the mPFC

To reversibly inactivate the prelimbic region of the mPFC during the retrieval phase of OPR testing at adulthood, we infused the GABA-A receptor agonist muscimol according to standard procedures[27]. The animals in two groups of these experiments were subjected to the early spatial experience as described for the Spatial-experience group of the main experiment. A third control group of animals was not exposed to any infantile experience. Between PD61-79 all animals were implanted bilaterally with guide cannula in the prelimbic region of the mPFC and then left undisturbed for at least 14 days. After the recovery period, the rats were randomly assigned to either the Muscimol or Saline group and subsequent preparations and testing on the OPR task followed the same procedures as for the Spatial-experience group, except that 18 min before OPR retrieval testing muscimol (HelloBio, Dunshaughlin, Ireland, 0.3 µg dissolved in 0.3 µL of 0.9% saline solution, per hemisphere) or an equivalent volume of saline solution was infused bilaterally over 3 min (at a rate of 0.1 µL/min) by an automated syringe pump (PHD ULTRA, Harvard Apparatus, Holliston, MA). For substance administration, a 33-gauge double-injection cannula (P1 Technologies, Roanoke, VA) was connected to

two 10 µL Hamilton microsyringes (Hamilton Company, Reno, NV) via a 1 m polyethylene tubing. The injection cannula protruded 1 mm beyond the tip of the guide cannula and was kept in the guide cannula for another 3 min to prevent backflow. Twelve minutes later, the animals were placed in the arena for OPR retrieval testing. Rats were perfused ~2 days after the experiments for histological confirmation of the infusion sites (Fig. S12). In two cases, the correct placement of the cannula could not be confirmed; respective data were discarded from the analyses.

## Surgery

Guide cannula were bilaterally implanted under general isoflurane anaesthesia (induction: 1–2%, maintenance: 0.8–1.2% in 0.35 l/min $O_2$). Preoperatively, fentanyl (0.005 mg/kg), midazolam (2 mg/kg) and medetomidine (0.15 mg/kg) were administered intraperitoneally. Rats were placed in the stereotaxic frame and the skull was exposed. A stainless steel double-guide cannula (5 mm long, 26 gauge, P1 Technologies) was implanted into the PL region of the mPFC (anterior–posterior (AP): 3.2 mm, mediolateral (ML): ±0.7 mm, relative to bregma, and dorsoventral (DV): −1.8 mm from the dura). The cannula was affixed to the skull with four bone screws and cold polymerizing dental resin. A double-dummy cannula (5 mm long, P1 Technologies) was inserted into the guide cannula and removed only for infusions.

## Statistical analyses

All statistical analyses were performed using SPSS software (IBM, Armonk, NY, USA). Generally, results are reported as means ± SEM. A $P < 0.05$ was considered significant. Analyses of the discrimination index (DI) and related behavioural control measures (total exploration time and total distance travelled) at adult OPR testing were first performed for the entire 5 min interval of the retrieval phase, and then focused on the first or first three min of this interval as this initial period is known to most sensitively reflect memory as assessed by the response to novelty[24]. With longer testing intervals, the novelty response systematically fades over time, thus adding noise to the memory assessment (e.g., refs. [70–72]). Statistical outliers were defined by a DI in the 1st minute of the retrieval phase exceeding ± 1.5 times the interquartile range (which correspond to the difference between the first and third quartile)[73], and excluded from analyses (1 case each in the Spatial-experience and Object-experience group, 2 cases in the Spatial-experience replication group). For the main experiment, a global analysis of variance (ANOVA) was performed with a Group factor representing the central experimental groups (Spatial-experience, Object-experience, No-experience, Stationary-experience, Context-change groups) and a repeated-measures Minute factor representing the 1st–5th minute of the retrieval phase. The Huynh-Feldt correction was applied when the sphericity assumption was violated. To specify the significant Group x Minute interaction ($F(9.801, 173.965) = 2.655$, $P = 0.005$) from this analysis, subsequent ANOVA were performed on subsets of groups. ANOVA on the effects of prelimbic mPFC-inhibition included a group factor Muscimol/Saline and were followed by post-hoc independent two-tailed t-test. Effects of age at spatial experience and sleep deprivation after infantile experience were tested in separate ANOVA including the Early childhood and Adolescence groups and the Sleep-deprivation group, respectively, in addition to the newly formed infantile Spatial-experience and No-experience groups. Early childhood and Adolescence groups were combined in these analyses to rule out potential biases resulting from unequal group sizes. Significant ANOVA main and interaction effects were followed by post-hoc pairwise t-tests (two-sided) and, in case of specific hypothesis testing, by planned contrasts. One-sample t-test was used to test whether DI values differed from chance level (zero).

c-Fos and COx activity values were first analysed by a global ANOVA including a Group factor (Spatial-experience,

Object-experience, No-experience, Long-term OPR, Context-change and Sleep-deprivation) and repeated measures Areas factor (PL, IL, CG, RSA, RSG, PAR, PRC, LEC, RE, RTN, CA1, CA3, DG). Significant Group x Areas interactions (Huynh-Feldt corrected) were followed by one-way ANOVAs including a Group factor (Spatial-experience, Object-experience and No-experience, or Spatial-experience, Context-change, and Long-term OPR; $n = 6$ for each group, except $n = 5$ for Context-change group), which were performed separately for each area. Significances were followed by post-hoc pairwise $t$-tests (two-sided). For an exploratory functional connectivity analysis, Pearson correlation coefficients were calculated (across individual rats) for c-Fos activity between all pairs of areas for each group separately. A positive correlation would be consistent with excitatory actions whereas a negative correlation would be consistent with inhibitory actions between the areas. Only correlation coefficients exceeding the criterion size of $r = \pm 0.815$ (corresponding to an uncorrected significance level of $P < 0.05$) were considered for a subsequent comparison of the total number of correlations exceeding the criterion size between conditions, using Fisher's exact test. Connectivity graphs were subsequently constructed using both c-Fos quantifications and correlation coefficients. The Igraph package (v1.2.4.2) in R (RStudio, Boston, MA) was used to visualize the networks.

### Reporting summary

Further information on research design is available in the Nature Portfolio Reporting Summary linked to this article.

## Data availability

All data needed to evaluate the conclusions in the paper are present in the paper and/or the Supplementary Material/Source Data. Source data are provided with this paper.

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

## Acknowledgements

We thank Ilona Sauter for technical assistance and Dr. Emily Coffey for proof reading. This research was supported by grants from the Deutsche Forschungsgemeinschaft to M.I. (DFG In 279/2-1) and to J.B. (DFG Bo854/18-1), the European Research Council to J.B. (ERC AdG 883098 SleepBalance), and from the Spanish Government (PSI2017-83893-R) to M.M. X.S. gratefully acknowledges funding from the China Scholarship Council (grant No. 201808080042). M.I. is supported by the Hertie Foundation (Hertie Network of Excellence in Clinical Neuroscience).

## Author contributions

Conceptualization: M.P.C., J.B., M.I. Methodology: M.P.C., J.B., M.I. Investigation: M.P.C., M.M., J.F., X.S., A.S. Visualization: M.P.C., J.F. Supervision: M.I., J.B. Writing—original draft: M.P.C., M.I. Writing—review and editing: J.B., J.F., M.I., M.P.C.

## Funding

## Competing interests

The authors declare that they have no competing interests.
