## [Peer Review File · Nature Communications]

Context memory formed in medial prefrontal cortex during infancy enhances learning in adulthoodREVIEWER COMMENTS

Reviewer #1 (Remarks to the Author):

This manuscript by Contreras and colleagues describes experiments that examined the influence of spatial learning in infancy on spatial learning in adulthood in rats. Young mice are exposed to objects that change spatial location and then subsequently tested in the object place recognition task in adulthood. The results demonstrate that the spatial experience in infancy results in the mice being able to perform the task in adulthood whereas the control groups that did not have this spatial experience in infancy were not able to perform the task (i.e., did not show object place recognition) in adulthood. This memory was abolished after inhibiting the medial prefrontal cortex and when the mice were sleep deprived after the spatial experiences in infancy.

The manuscript describes novel, impactful results that I believe will be of wide interest. It is clearly written and the data and experimental designs are clearly communicated in the Figures. I have only minor comments that I feel would improve the communication of the results.

1. Statistics – For the most part the statistics are appropriate and comprehensive and support the conclusions however there are a few curiosities. As it is the case that the behavioural phenotypes are most distinguishable in the first minute of testing it would be helpful to see this consistently reported and displayed in the graphs of the main text. This is done now for many of the experiments however the 1st 3 minutes shown in Fig 4. The group by minute interactions indicate that the expression of the discrimination changes over time differently in the groups and I think it would be helpful to report these stats and plot the data for each of the minutes in the supplemental material. This is done now for some but not all the experiments. Additionally, I am not convinced that combining Early childhood and Adolescence groups is an appropriate or necessary approach for dealing with unequal group sizes.

2. Control analyses that account for group differences in exploration time and distance traveled are described in the S1 figure legend. I think that it is appropriate to include these in the supplemental material, but the result could be cited in the main text and described in a separate section of the supplemental material.

3. The results displayed in Figure 3 show increase Fos relative to a homecage control in many regions. This strikes me as non-specific and could just be a general arousal response. Inclusion of an object exp group would point to specificity.

4. Sleep deprivation of pups could be considered a stress exposure following learning so the effect on subsequent memory may not be specific to sleep deprivation.

5. As the rats show a familiarity preference (though not a novelty preference) when encoding occurs in the infantile stage and remote retrieval occurs in adulthood the conclusion that rats do not retain an episodic representation over this long period of time (line 127) is not supported.

Reviewer #2 (Remarks to the Author):

The outstanding questions targetted by this study can be summarized as: why are infantile memories, that are widely thought to affect adult behaviour, ostensibly forgotten? Are they really forgotten? What types of memories are they? And how can this be explained neurally? These are really interesting questions. The experiments are well-described and the authors' conclusions are clear.

The case is built up in step-by-step experimental narrative, which is logical and easy to follow. I thought it would make sense, therefore, to review the experiments in the same step-by-step manner. I find as I go through that some questions arise and some loose ends seem to require tying up; perhaps these comments will help the authors address these issues via changes to the

MS, or by further experiments. The headings below are taken from the Results section of the MS.

"Infantile spatial experience enhances OPR performance at adulthood":

In this first experiment it is shown that pre-exposure of the spatial-exposure (S-E) group of rats to the location recognition OPR task makes them better later on the OPR task. Exposure to a different object recognition task (object exposure group, O-E) does not. The O-E group is a very nice control for the S-E group. The results of previous task performance facilitating subsequent task performance, especially in the same context, are expected, so at this early stage there are no new or surprising findings telling us anything particularly new; this is the foundational experiment that subsequent experiments will build on. Having said that, a new question is raised in that we don't yet know exactly why the pre-exposed rats were better. A simple explanation is that rats exposed to OPR have encoded the spatial environment better than the O-E group because they had to use the information to solve the task. Perhaps the O-E group was distracted by the changing objects and so did not encode the environment as well as the other groups. It is concluded that "Overall, these results demonstrate that discrete and short-lasting spatial experience during infancy distinctly impacts the capability to form stable spatial representations in adulthood." So it sounds like the interpretation is that what is remembered is the "spatial representation" of the environment. However, both S-E and O-E groups had the same experience of the environment (it's a very good control, in that sense), so I'm not sure that makes sense, at least at this early stage in the story (unless one assumes increased encoding of space in the S-E groups due to mechanisms like those suggested above). In any case, so far this says nothing about infantile amnesia, only that exposure to a task makes rats better at that task in the future. I guess a control group is needed in which the pre-exposure is done in non-infant animals. We get that experiment later in the study.

I would add that Figure S1 shows that there are many differences between the groups in terms of total object exploration and distance travelled in the arena. Normally in these types of experiments these measures do not differ, as such differences can fatally confound interpretation of the data. Indeed, I'm not actually sure whether these measures are from the OPR pre-exposure or the OPR test. Even more important is the exploration during pre-exposure, because that effects the amount of encoding going on, which effects the extent to which memory will be expressed later. So I am left with the feeling of being unsure about the veracity of the main effect. Perhaps a discussion of why these differences should not affect interpretation would help assuage these concerns.

"Adult rats show no episodic memory for spatial exposures during infancy"

Here, the question of what is remembered is addressed, and the conclusion is that what is remembered is not an episodic memory for the 'spatial experience' or 'spatial exposure' (meaning the spatial environment, I guess), but something else. The evidence for this is that a single exposure to the sample locations – not the whole task done four times, as in the previous experiment – resulted in no evidence of memory for the object locations. The authors emphasize that this is the "strictest test" of episodic memory. But that means the way it's tested biases the result toward the eventual conclusion. I can't see how it tells us much about what is remembered in the previous experiment, as it doesn't at all resemble the previous experiment. Truth is, I'm not sure exactly why this experiment was done.

And in fact, some evidence was found for intact memory (significant preference for the novel location, which indicates memory). However this is interpreted as a "rudimentary" type of memory, rather than episodic. No reason is given for this interpretation.

So at the end of this experiment, we don't yet know what is and is not remembered in the initial main experiment. At this point there are still no findings that are shown to be specific to infantile amnesia.

"Effects of infantile spatial experience on the adult rat's learning capability are context-dependent"

The authors "hypothesized that infantile spatial experience improved adult spatial memory capabilities via inducing a more abstract schema-like representation which in theory is context-

independent." Thus, in this experiment two groups of rats were tested, one that had OPR task exposure in context 1, and OPR testing in context 2, and another that had OPR task and testing in the same context (the rats in the first experiment). The manipulation of context extended to the identity of the experimenter, which is good and thorough. Only when the context matched did rats benefit from pre-exposure to the task. This is what one would expect, based on the idea (see above) that it is encoding of the spatial environment (context) that helps the pre-exposed animals perform better. The authors conclude "These results support the notion that schema-like spatial memories formed during infancy can also integrate contextual elements of the infantile experience." This is what would be predicted, and so doesn't really say much new, except for the statement that this contextual/spatial memory is "schema-like" (in the Discussion it is described as "abstract"). Unfortunately, so far in these experiments I don't see any evidence that the memories are schema-like or abstract. To test that, experiments would need to be specifically designed to test whether memories are "schema-like" or not. Indeed the hypothesis above says that a schema-like representation would be context-independent. But the facilitation by pre-exposure is context-dependent, and so isn't the conclusion the opposite, that the memory is not schema-like? I am getting a little confused at this point.

At this point there are still no findings that are shown to be specific to infantile amnesia.

"Infantile spatial experience forms memory in the prelimbic region of the mPFC"

In this experiment, the authors test whether whatever-memory-is-responsible-for-improved-performance-in-the-S-E-group is transferred from hippocampus to prefrontal cortex in a systems-level-consolidation-type manner. This is done using c-fos imaging, comparing activity in hippocampus and prefrontal cortex. First it needs to be acknowledged that the results will be hard to interpret because the comparison between groups is confounded by performance: the S-E groups perform the task well prior to c-fos analysis, and the other groups don't. Putting that aside for the moment, no between-group differences in c-fos were found in the hippocampus. Instead, differences were found in the prefrontal cortex, specifically in the prelimbic cortex where c-fos activity in S-E group was greater than in the other 2 groups. This agrees with the expectations of the study. However this finding was accompanied by a number of other unexpected (I think) changes such as greater activity in the PAR, PRC, and LEC in the No-exposure group compared to the S-E group. I have no good explanation for these unexpected findings, but it does make me wonder whether we should be focussing and putting our trust in the one finding that agrees with the expectations of the study, when there are all these other changes we don't understand. Having said that, the prelimbic finding does seem sound as it was subsequently replicated in other ways.

As mentioned above, there has been no convincing evidence presented so far that the facilitating memories are schema-like, or abstract, or non-episodic. Throughout this section a lot of conclusions are made about the nature of the memory, based on the structures active according to c-fos etc. But structures are not memories, and although the system-level-consolidation-episodic-to-schema type model is believed by many, it is also heavily contested. So we really can't be making conclusions about the nature of a memory on the basis of the activity of brain structures, even if the data were clear, unconfounded, and unproblematic.

And at this point there are still no findings that are shown to be specific to infantile amnesia.

Also I'd prefer a causal experiment to test the necessity of the putative brain structures. We get that next.

"Inhibiting the prelimbic region of mPFC at adult OPR testing abolishes benefits from early spatial experience"

Prelimbic cortex was silenced prior to the retrieval phase of the OPR task, during the test (not the pre-exposure). Performance was impaired in the muscimol group compared to the saline group (both groups had spatial pre-exposure). It is concluded that "These results indicate that the PL mPFC is critically involved in mediating the improving effects of infantile spatial experiences on the adult rat's spatial learning capabilities." However the effect could be just muscimol impairing the task, rather than anything to do with the effects of pre-exposure. A control for this might be

something like muscimol and saline in rats that performed the task well, but that had not been pre-exposed.

And at this point there are still no findings that are shown to be specific to infantile amnesia. But wait ...

"Effects of early spatial experience on adult OPR performance depend on developmental age"

Here spatial exposure is given to rats at three different ages -- infancy, early childhood and adolescence -- and OPR tested in adulthood. The infancy group performed well. Rats exposed in early childhood or adolescence did not perform well, suggesting that the initial finding was specific to infantile amnesia. It's great that an experiment was included to test whether the main finding was specific to infantile amnesia. However some features of the data make me uneasy. For example, the Infancy group has an N of 27, and the Early Childhood group has an N of 7. Putting aside the statistical problem of uneven sample sizes, can we really trust the conclusions when they depend on finding an effect in the highly-powered group, and no effect in the low-powered group?

In any case what I don't understand is why this would happen, even according to the authors' favored interpretation. The story here is that the infants don't form episodic memories, only "rudimentary", "abstract" ones. The older groups would have episodic memories of the experience. So why does "rudimentary" memory facilitate subsequent performance, but not episodic memory? This is counter-intuitive, and needs to be explained.

"Effects of early spatial experience on adult OPR memory depend on post-experience sleep"

In this experiment, sleep deprivation in infancy affected performance on test in adulthood. But we don't know from this experiment whether sleep deprivation affected the facilitatory effects of pre-exposure, or just performance of the task in adulthood.

Summary

I'll say again that this is well-written description of studies targeting some really interesting questions about infantile amnesia. However as is clear from the above, I think there are a number of outstanding issues that need to be addressed, either in revisions to the MS or more likely with additional experiments and control groups. Only one experiment (out of 8 or 10) really addresses infantile amnesia per se and includes non-infant rats as controls. Other experiments want to talk about the effect of pre-exposure but have no non-pre-exposure group. Other experiments have peculiarities in the data or possible confounds (total exploration, side effects of sleep) that compromise unequivocal interpretation. The major conclusion seems to be about the nature of the facilitating memory being schema-like or abstract, and I see no convincing evidence for that conclusion; indeed some evidence seems to indicate the opposite.

Some further, more minor comments

In the Introduction, it is made clear that a number of studies have already looked at this problem and have offered some answers. That suggests that perhaps this study is not so novel. So what is new? What is new, according to the Introduction, is that previous studies all used aversive methodology to study 'emotional' memories, whereas as the present study does not. To the non-expert, this might seem like a possibly trivial tweak in methodology. Rather than just state that something was done differently, I think it needs to be explained why the shift from aversive to non-aversive methodology is important. How might the previous studies have given us the wrong answer because of the methodology used? Is it due to possible confounds, for example the effects of stress and stress hormones? I think the rationale and importance of this novel aspect of the study could be expanded and made more explicit.

The data in Fig 1B show that only the S-E group performed well on the OPR task; the other groups attained scores that are for the most part no different from chance. The authors say this low performance "replicates previous studies" but cite only one study from their own lab. These tasks have been used in a huge number of studies, and performance this poor is highly unusual. This

does not negate the fact that the S-E group did perform well, and indeed one might have intentionally designed the study using parameters that generated low baselines so that improvements could be more easily seen (for example, by using a 24-hour delay, or more similar locations). Nevertheless, the rats' performance is so poor at a 3-hour delay that I can't help but be concerned about how the task is being run. This concern is amplified by the huge amount of variability in the data; the whole range of possible scores is represented, from -1 to 1. This is also unusual for this type of task when run properly. In any case, I think readers will want to be convinced that the methodology is sound and so this issue should be discussed and cited more openly and accurately, and some possible reasons for the poor performance should ideally be given.

The citations are often focussed on a few currently popular authors (newcomers to this area, as well) and reviews in 'glamor' publications. So this is just a reminder to the authors that we always want to cite the earliest, primary sources.

Reviewer #1 (Remarks to the Author):

This manuscript by Contreras and colleagues describes experiments that examined the influence of spatial learning in infancy on spatial learning in adulthood in rats. Young mice are exposed to objects that change spatial location and then subsequently tested in the object place recognition task in adulthood. The results demonstrate that the spatial experience in infancy results in the mice being able to perform the task in adulthood whereas the control groups that did not have this spatial experience in infancy were not able to perform the task (i.e., did not show object place recognition) in adulthood. This memory was abolished after inhibiting the medial prefrontal cortex and when the mice were sleep deprived after the spatial experiences in infancy.

The manuscript describes novel, impactful results that I believe will be of wide interest. It is clearly written and the data and experimental designs are clearly communicated in the Figures. I have only minor comments that I feel would improve the communication of the results.

Authors' response: Many thanks! We are very grateful for this overall positive judgement of our work.

Reviewer's comment 1: Statistics – For the most part the statistics are appropriate and comprehensive and support the conclusions however there are a few curiosities. As it is the case that the behavioural phenotypes are most distinguishable in the first minute of testing it would be helpful to see this consistently reported and displayed in the graphs of the main text. This is done now for many of the experiments however the 1st 3 minutes shown in Fig 4. The group by minute interactions indicate that the expression of the discrimination changes over time differently in the groups and I think it would be helpful to report these stats and plot the data for each of the minutes in the supplemental material. This is done now for some but not all the experiments. Additionally, I am not convinced that combining Early childhood and Adolescence groups is an appropriate or necessary approach for dealing with unequal group sizes.

Authors' response: We thank the reviewer for asking to clarify our data in Fig. 4. There was no significant group-by-minute interaction here but only a main effect across the 1st and first 3 min. We have now added a graph showing the discrimination index across the 5 min to the Supplemental material for the data in Figure 4, i.e. the new supplementary figure S10, and added to the legend that the Group x Minutes interaction effects was not significant.

As to combining Early childhood and Adolescence groups, we agree that the pooling of data is not absolutely necessary, However, given that data from these two groups were closely comparable pooling represents an appropriate statistical measure to enhance power of the comparison with the infantile group. With regard to the significant Group x Minutes interaction effect we clarified here that the difference between age groups concentrated on the first min (p. 18):

“...OPR performance at adulthood testing was indistinguishable between the Early childhood and Adolescence groups ($t(26) = -0.053, -0.112, 0.033, 0.231$ and -0.074 , all $P > 0.819$ for

pairwise comparison per minute), but was significantly worse than that of the infantile Spatial-experience group with this group difference being strongest for the 1st min of the retrieval phase ($F(1.883, 212) = 5.413$, $P = 0.007$ and $P = 0.36$, for Group x Minutes interaction and Group main effect, respectively, in an ANOVA with pooled Early childhood and Adolescence groups, Fig. 5A).”

The new supplementary figure S10 is shown below:

Figure S10. Adult OPR performance for the entire 5-minute retrieval phase for the Saline and Muscimol groups. Mean±SEM discrimination ratios separately during the first 1, 2, 3, 4 and entire 5 min of the retrieval phase at adulthood OPR testing, for the Saline ($n = 6$) and Muscimol ($n = 6$) groups (dot plots overlaid). # $P < 0.05$, one sampled t-test against chance level. Differences between the groups for the first 3 min were significant ($F(1, 34) = 8.4$, $P < 0.01$, for main effect of Group, $P = 0.84$ for Group x Minutes interaction, see main text).

Reviewer’s comment 2: Control analyses that account for group differences in exploration time and distance traveled are described in the S1 figure legend. I think that it is appropriate to include these in the supplemental material, but the result could be cited in the main text and described in a separate section of the supplemental material.

Authors’ response: Yes, as suggested, we have now added respective results to the main text, and extended the description of these data in the Supplemental material (please, note original fig. S1 is now fig. S2 to keep the order of figures according their mentioning in the text):
The changes to the main text are as follows (Results section, p. 4-6):

“...The enhancement was largest during the first minute of the retrieval phase which is typically most sensitive to the memory effect²⁴ ($F(2, 51) = 4.464$, $P = 0.017$, for main effect of Group, see Fig. 1B for results from pairwise statistical comparisons). In fact, in the 1st min of retrieval testing. Analysis of behavioral control parameters revealed that Spatial-experience and Object-experience groups were closely comparable with respect to total exploration time (fig. S2A). However, rats of the Spatial-experience group travelled a slightly greater distance than the rats of the Object-experience group ($t(32) = -2.055$, $P = 0.048$), possibly reflecting general arousing effects on locomotion resulting from stimulation specifically of spatial systems during infancy. We excluded travelled distance as a factor possibly confounding memory performance in additional analyses including distance travelled as covariate, which confirmed significance for the difference in discrimination indexes between the Spatial- and Object-experience groups ($F(1, 31) = 8.416$, $P = 0.007$).

Control analyses of the behavior at the four exposures during infancy...”

In the Supplementary material we extended the text to fig. S2A as follows:

„...Note, in (A), Spatial-experience and Object-experience groups were closely comparable with respect to total exploration time. However, rats of the Spatial-experience group travelled a slightly greater distance than the rats of the Object-experience group ($t(32) = -2.055, P = 0.048$), possibly reflecting general arousing effects on locomotion resulting from stimulation specifically of spatial systems during infancy. To exclude a confounding influence of locomotion on OPR retrieval performance, we ran additional control analyses including distance travelled at retrieval as covariate, which confirmed significance for the difference in discrimination indexes between the Spatial- and Object-experience groups ($F(1, 31) = 8.416, P = 0.007$). In (B) diminished...”

Reviewer’s comment 3: The results displayed in Figure 3 show increase Fos relative to a homecage control in many regions. This strikes me as non-specific and could just be a general arousal response. Inclusion of an object exp group would point to specificity.

Authors’ response: Yes, the reviewer is absolutely right in that the comparison with homecage control group data (in Fig. 3) does not prove that the activation of mPFC areas observed at encoding in the Spatial-experience group is specifically related to the encoding of the spatial experience in the pups. Indeed, we performed the experiments shown in Fig. 3 after the main

experiments had revealed enhanced prelimbic mPFC activation at OPR testing in adulthood for the rats of the Spatial-experience group. The question was simply, whether activity in the prelimbic region of the mPFC is already enhanced at encoding of spatial experience during infancy. We did not intend to address the more far-reaching question whether encoding of the spatial experience at infancy differs from encoding of the object-experience. Accordingly, we have now more carefully phrased the conclusion to be drawn from this data. (Please, note also our response to the related comment 1, by Reviewer #2):

(p. 12):

“...($F(12,120) = 2.750, P = 0.002$ for ANOVA Group x Areas interaction, see Fig. 3B for pairwise comparisons). Overall, the observed engagement of the prelimbic mPFC at encoding of the spatial experience during infancy is consistent with the view that the representations used to enhance OPR performance of the Spatial-experience group at adulthood, were rather quickly formed already during infancy, although these additional experiments in infant rats did not aim to specifically dissociate encoding of spatial experience (from object experience)...”

Reviewer’s comment 4: Sleep deprivation of pups could be considered a stress exposure following learning so the effect on subsequent memory may not be specific to sleep deprivation.

Authors’s response: Yes, although unlikely, we can presently not entirely exclude that the deprivation of the pups induced stress that confounded the effects of mere wakefulness. We have extended the discussion of this issue as follows (p. 19):

“...experience group ($t(10) = 2.768, P = 0.020$, for pairwise comparison between groups).

It is unlikely that these effects of sleep deprivation were confounded by stress capable to induce general adverse effects on brain development, as sleep deprivation of the pups was performed very cautiously using gentle handling in the presence of the littermates and close to the mother that could be seen and smelt by the pup. Moreover, duration of sleep deprivation was relatively short (1.5 h), and none of the pups showed any behavioral signs of stress or fear during sleep deprivation. The use of similar deprivation procedures in adult rats remain without any effect on stress hormones levels like corticosterone (e.g., Melo and Ehrlich, 2016). Thus, assuming that the sleep deprivation procedure did not induce substantial side effects, our findings support the conclusion that the beneficial effect of infant spatial experience on adult spatial learning requires sleep to occur after the infant experience.”

Reviewer’s comment 5: As the rats show a familiarity preference (though not a novelty preference) when encoding occurs in the infantile stage and remote retrieval occurs in adulthood the conclusion that rats do not retain an episodic representation over this long period of time (line 127) is not supported.

Authors’s response: We are not sure whether we fully understand the reviewer’s point here. In the Long-term OPR condition the pups encoded the task on PD24, whereas we see in our foregoing experiments (Contreras et. al., 2019) familiarity preference only at a distinctly earlier age (PD18). Moreover, conceptually, the presence of a specific episodic memory representation may be considered independent of its behavioral assessment, i.e., one and the

same object configuration encoded at PD18 maybe behaviorally retrieved via familiarity preference, when tested at PD18, but retrieved via novelty preference when tested at PD24 or later. As this conceptual consideration, indeed, needs still empirical confirmation, we have rephrased the respective sentence more cautiously (p. 9). We also removed conclusions regarding the Long-term OPR condition from the Abstract.

(p. 9) “...In line with other studies^{6, 26, 27}, this result suggests that the rats did not retain a full-blown and detailed episodic representation - of the configuration of objects within its specific spatial context - over such a long time, thus...”

Reviewer #2 (Remarks to the Author):

The outstanding questions targeted by this study can be summarized as: why are infantile memories, that are widely thought to affect adult behaviour, ostensibly forgotten? Are they really forgotten? What types of memories are they? And how can this be explained neurally? These are really interesting questions. The experiments are well-described and the authors' conclusions are clear.

The case is built up in step-by-step experimental narrative, which is logical and easy to follow. I thought it would make sense, therefore, to review the experiments in the same step-by-step manner. I find as I go through that some questions arise and some loose ends seem to require tying up; perhaps these comments will help the authors address these issues via changes to the MS, or by further experiments. The headings below are taken from the Results section of the MS.

Reviewer's comment 1: "Infantile spatial experience enhances OPR performance at adulthood":

In this first experiment it is shown that pre-exposure of the spatial-exposure (S-E) group of rats to the location recognition OPR task makes them better later on the OPR task. Exposure to a different object recognition task (object exposure group, O-E) does not. The O-E group is a very nice control for the S-E group. The results of previous task performance facilitating subsequent task performance, especially in the same context, are expected, so at this early stage there are no new or surprising findings telling us anything particularly new; this is the foundational experiment that subsequent experiments will build on. Having said that, a new question is raised in that we don't yet know exactly why the pre-exposed rats were better. A simple explanation is that rats exposed to OPR have encoded the spatial environment better than the O-E group because they had to use the information to solve the task. Perhaps the O-E group was distracted by the changing objects and so did not encode the environment as well as the other groups. It is concluded that "Overall, these results demonstrate that discrete and short-lasting spatial experience during infancy distinctly impacts the capability to form stable spatial representations in adulthood." So it sounds like the interpretation is that what is remembered is the "spatial representation" of the environment. However, both S-E and O-E groups had the same experience of the environment (it's a very good control, in that sense), so I'm not sure that makes sense, at least at this early stage in the story (unless one assumes increased encoding of space in the S-E groups due to mechanisms like those suggested above). In any case, so far this says nothing about infantile amnesia, only that exposure to a task makes rats better at that task in the future. I guess a control group is needed in which the pre-exposure is done in non-infant animals. We get that experiment later in the study.

Authors' response: We are very grateful to the reviewer for this valuable comment which indeed helped to substantially advance the interpretation of our findings. As we see the point made by the reviewer, he/she argues that the rats of our Object-exposure (O-E) group performed worse than the Spatial-exposure group at adult OPR testing because they simply did not encode the important spatial information: ... "Perhaps the O-E group was distracted by the change in objects and so did not encode the environment as well as the other groups". Although, initially, we found this explanation unlikely, we decided to run an additional control experiment

with the pups exposed during infancy to the same object configurations as the Object-experience group except that the objects at the first and second visit of an exposure on the same day remained the same (see Figure 1, below). We termed this control group “Stationary-experience” group because within each of the four exposures during infancy, both objects and their spatial configuration *did not change*. Contrary to our expectation, at adult OPR testing these rats performed as well as our Spatial-experience group, and significantly better than the rats of the Object-experience and No-experience control groups. Indeed, although unexpected, this finding helps to more precisely answer the key question (also asked by this Reviewer), i.e., what is the information (at the exposures during infancy) which the rats use at adulthood OPR testing to improve their performance. The answer is: Spatial context information at the infancy exposures alone is sufficient to produce the enhanced OPR performance at adulthood. To integrate these experiments and findings into the ms and to accordingly adapt the interpretation of our findings, the following changes have been made to the Results and Discussion section as well as and to the Abstract:

Results - Here we introduced a new paragraph (p. 7) and rearranged the two following paragraphs:

“Context information at infantile exposures is sufficient for enhancing the adult rat’s OPR performance

The infantile exposures in our Spatial-experience group comprised changes in the configuration of the two objects (from the first to the second visit of the arena) with the environmental arena context remaining the same throughout the four exposures. The arena context for the Spatial-experience group being the same as for the Object-experience group suggests that the rats of the Spatial-experience group relied on the experienced changes in the spatial configuration of the two objects, rather than on contextual information, to enhance OPR performance at testing in adulthood. Yet, it could also be argued that in the Object-experience group the change in the object during the infantile exposures distracted these pups from encoding context information potentially relevant for their OPR performance during adulthood. Hence, to further specify the information the rats of the Spatial-experience group used at infantile exposures to enhance their adult OPR performance, we examined an additional group of rats (**Stationary-experience**). Rats of this group were subjected to basically the same procedures as the Object-experience group, except that the objects during the two visits of the arena remained the same (Figure 1C). These exposures, thus, did not only lack any change in the spatial configuration of the two objects but also the potentially distracting effect of a change in the objects. Notably, at OPR testing during adulthood, the Stationary-experience group profited from the infantile exposures in the same way as the Spatial-experience group, i.e., OPR performance of these rats was significantly better than that of the Object-experience and No-experience control groups ($F(2, 46) = 3.409, P = 0.042$, for the group main effect across these 3 groups), and closely comparable with that of the Spatial-experience group ($P = 0.816$, for the pairwise comparison between these groups). Similar to the Spatial-experience group, discrimination preference in the rats of the Stationary-experience group appeared to remain at a high level throughout the 5-min retrieval phase (fig. S4). Indeed, these findings, suggest that, for enhancing OPR performance at adulthood, the pups need to only encode the spatial context information at the infantile exposures. (Note that encoding the context implicates an event experienced which in the Stationary-experience group was represented by the different objects in at each infantile

exposure.) On the other hand, the exposure to a change in the spatial configuration of the two objects – as experienced by the pups of the Spatial-experience group - is not required.

Effects of infantile spatial experience on the adult rat's learning capability are context-dependent

If the enhanced OPR performance at adulthood in rats of the Spatial-experience group was primarily owed to the memorization of environmental context information encoded during the exposures during infancy, we would expect that the enhancement in OPR performance in these rats is restricted to the same context as that during the infantile exposures. To test this hypothesis, we compared the Spatial-experience rats which performed the OPR task at adulthood in a very similar context as that used for inducing spatial experience during infancy (same experimenter, same distal cues) with a control group of Spatial-experience rats which performed the OPR task at adulthood in an entirely different context (Context-change group; Fig. 1C). We found that the Context-change group did not profit from the infantile spatial experience ($F(1, 27) = 7.490, P = 0.011$, for the difference between this group and the Spatial-experience group tested in the original context). In fact, the Context-change rats did not perform above chance level at any minute of the OPR retrieval phase at adulthood (all $t(11) > -0.759$ and $P > 0.160$, fig. S4). These results corroborate the view that memories mainly containing contextual information of the infantile exposures helped the rats of the Spatial-experience group to enhance their OPR performance at adulthood testing.

Adult rats show no episodic memory for spatial exposures during infancy

In a further control experiment, we asked whether the enhanced spatial memory capabilities in the Spatial-experience group were perhaps a direct consequence of an episodic memory that was formed for the individual exposures of the spatial experience during infancy and persisted into adulthood. To answer this question”

Figures see below

Figure 1: Effect of infantile spatial experience on adult OPR performance. (A) General procedure: During infancy, pups of the Spatial-experience group ($n = 17$, red) were placed in an arena with two identical objects for 5 min and, after a 5-min break, re-entered to the arena but this time, one of the objects was displaced to a new location. Different spatial configurations and objects were used at the four arena visits, on PD18, 20, 22, and 24. For the Object-experience group ($n = 17$ rats, green), instead of a change in object location, one of the objects was replaced by another, in the second 5-min period. The No-experience group ($n = 18$ rats, grey) had no arena visits during infancy. At adulthood (~PD80), all groups were tested on a classical object-place recognition (OPR) task with a 3-hour delay between encoding and retrieval testing. (B) OPR memory (mean \pm SEM discrimination ratios during 1st min of retrieval phase, dot plots overlaid) at adulthood testing. Only rats with spatial experience during infancy displayed significant OPR memory. (C) Grey shaded - Procedure of additional control experiments (right). For the rats of the Stationary-experience group ($n = 12$ rats, yellow) both the objects and their spatial configuration remained unchanged at the two visits of each infantile exposure. Procedures for the Context-change group ($n = 12$ rats, purple) were the same as for the Spatial-experience group, except that OPR testing at adulthood was performed in an entirely different context. Whereas the Stationary-experience group showed the same enhanced OPR performance as the Spatial-experience group, the Context-change groups did not show significant OPR memory (bottom). ### $P < 0.001$ for one-sample t-test against chance level; * $P < 0.05$ and ** $P < 0.01$ for pairwise comparisons (two-sided t-tests) between experimental groups. (see figs. S1 and S4 for discrimination ratios for entire 5-min retrieval phase).

Figure S4. OPR memory for the entire 5-minute retrieval phase in the Stationary-experience and Context-change groups. Mean±SEM discrimination ratios for the Stationary-experience (n = 12 rats, yellow bars) and the Context-change (n = 12 rats, purple bars) control groups shown separately during the first 1, 2, 3, 4 and entire 5 min of the retrieval phase, in comparison with the Spatial-experience group (n = 17; red bars; dot plots overlaid). For the Stationary-experience group, objects and their spatial configuration remained unchanged at the two visits of each infantile exposure. Procedures for the Context-change group were the same as for the Spatial-experience group, except that OPR testing at adulthood was performed in an entirely different context. ## $P < 0.01$ and ### $P < 0.001$ for one-sample t-test against chance level. * $P < 0.05$, and ** $P < 0.01$ for pairwise comparisons (two-sided t-tests) between experimental groups. ($F(1, 27) = 0.064$, $P = 0.803$ and $F(1, 27) = 31.943$, $P = 0.001$, for group main effect in ANOVA comparing the Spatial-experience group with the Stationary-experience and Context-change groups, respectively).

Discussion – Here, we mainly changed the first paragraph (summary of main findings, p. 20), and those parts concerning the kind/content of representation formed during the infantile exposures, which produced the improvement in spatial memory capability at adulthood (starting with p. 21).

Summary of main findings (p. 20):

“We present a novel approach that seeks to characterize the influence of discrete non-emotional spatial experience during infancy on spatial learning in adulthood. We find that a seemingly insignificant event, i.e., a change in the spatial configuration of two objects the rat is exposed to in an experimental arena during its infancy for a few times - overall no more than 20 min and in the absence of any rewarding or aversive stimulation - distinctly impacts learning behavior and related brain organization during adulthood: This means at adulthood, the rats displayed enhanced capabilities to form spatial memories. Notably, the adult rats’ memory capabilities were similarly enhanced when during infancy the rats were exposed to the same two objects presented also twice on each exposure but, in the *absence* of a configurational change, suggesting that the memories producing the adult rats’ enhanced spatial memory capability were based on contextual information rather than the experience of the change in the object configuration. The enhancement in spatial memory performance in adulthood was specifically related to the use of representational systems residing in the prelimbic region of the medial prefrontal cortex (mPFC)...”

Interpretation regarding the content of memory formed at infancy (p. 21ff):

“ What is the content of the memory formed during infantile spatial experience, that enhances memory capabilities at adulthood? To address this core question of our behavioral experiments we examined adult rats’ OPR performance in two additional conditions, i.e., the Stationary-experience and the Context-change conditions. The infantile exposures in the Stationary-experience group lacked the experience of a change in the spatial configuration of the two objects but were otherwise identical to those of the Spatial-experience group. Importantly, the exposures took place in the very same environmental arena context as that used for the pups of the Spatial-experience group. Thus, the rats of the Stationary-experience group benefitting (at OPR testing in adulthood) from the infantile exposures to the same degree as the Spatial-experience group strongly suggests that the enhanced OPR performance these rats showed in adulthood, resulted from memories formed of the arena context, whereas the mere experience of a change in the configuration between the objects seems to be of secondary relevance. This view that rats of the Spatial-experience group relied on memory of contextual information about the infantile exposures at adult OPR testing, is corroborated by our findings in the Context-change group. These animals did not show an enhanced OPR performance at adult testing although during infancy they had been exposed to the same spatial experience as the Spatial-experience group. Yet, OPR testing at adulthood took place in a context entirely different from that experienced during the exposures at infancy, i.e., in a different room with different distal cues, a different floor and with a different experimenter. Accordingly, we assume that the rats of the Spatial-experience group benefited from an enduring memory of such spatial and social features of the environmental context that belonged to the infantile exposures and acted as reminders when they were tested on the OPR task as adults.

Although our behavioral findings conclusively support the view that the improved adult OPR performance of the Spatial-experience group was based on contextual memories of the infantile exposures, this explanation is difficult to reconcile with the findings in the Object-experience group: This group did not benefit from the infantile exposures at adult OPR testing, although their exposures during infancy took place in the same environmental context as in the Spatial experience group. Instead of a change in the spatial configuration of two identical objects, rats of the Object-experience group were exposed to a change in one of the two objects, i.e., at the second arena visit of each exposure one of the objects was replaced by a novel object. Although we did not find behavioral signs of an increased interest in the objects during the infantile exposures in these pups (fig. S3), one explanation for their lack of benefit at adult OPR testing could be that during their infantile exposures the pups were distracted by the novel objects from encoding the relevant context information. Indeed, the conditions are reminiscent to those found in 15-20 months old human infants who did not discriminate the spatial room context while searching for toys hidden in boxes but, surprisingly, showed the ability to disambiguate the boxes according to the room context when the toy cues were absent (Newcombe et. al., 2014). Nevertheless, the interpretation that the novel objects distracted the rat pups of our Object-experience group from encoding spatial context remains tentative and needs to be scrutinized in further experiments.

Our finding indicating that benefits for spatial capabilities in adulthood originate from memory of contextual information formed during infantile experiences well fits the observation that the memory recall in these early years of life shows a distinctly greater context-dependency

than in adolescence and adulthood⁵⁶. Of note, rat pups have been shown to encode contextual cues into mPFC regions including the prelimbic region, from early on (i.e., on PD16)⁵⁷ and, here, c-Fos activity in the same mPFC region was enhanced above levels in home cage controls already after encoding of the fourth spatial exposure, suggesting that contextual information forms part of the supraordinate representation mediating the enhancing effects of infantile spatial experience on adult learning capabilities. A preferential formation of persisting contextual memories in mPFC networks might be advantageous as such memories might serve as reference frame, not only scaffolding the recall of multiple episodes experienced in the same or similar context (e.g., Robin and Moscovitch, 2017; Burgess et. al., 2002; Ekstrom and Ranganath, 2018) but also effectively guiding future behavior in such context, thereby supporting - in a context-dependent manner - learning processes like those seen in our Spatial-experience group.

While in the mature brain, the scaffolding of detailed memory recall by spatial context information has been commonly linked to hippocampal function (e.g., Hassabis and Maguire, 2007), the hippocampus ...”

Title and Abstract – We have slightly modified the title to:

“Context memory formed in medial prefrontal cortex during infancy enhances learning in adulthood”

The Abstract now reads as follows:

“Adult behavior is commonly thought to be shaped by early-life experience, although episodes experienced during infancy appear to be forgotten. Exposing rats during infancy to discrete spatial experience (configurational changes of objects in an arena) we show that these rats in adulthood are significantly better at forming a spatial memory than control rats without such infantile experience. We moreover show that the adult rats’ improved spatial memory capability is mainly based on memory for context information during the infantile experiences. Infantile spatial experience increased c-Fos activity at memory testing during adulthood in the prelimbic medial prefrontal cortex (mPFC), but not in the hippocampus. Inhibiting prelimbic mPFC at testing during adulthood abolished the enhancing effect of infantile spatial experience on learning. Adult spatial memory capability only benefitted from spatial experience occurring during the sensitive period of infancy, but not when occurring later during childhood, and when sleep followed the infantile experience. In conclusion, the infantile brain, by a sleep-dependent mechanism, favors consolidation of memory for the context in which episodes are experienced. These representations comprise mPFC regions and context-dependently facilitate learning in adulthood.”

Reviewer’s comment 2: I would add that Figure S1 shows that there are many differences between the groups in terms of total object exploration and distance travelled in the arena. Normally in these types of experiments these measures do not differ, as such differences can fatally confound interpretation of the data. Indeed, I’m not actually sure whether these measures are from the OPR pre-exposure or the OPR test. Even more important is the exploration during pre-exposure, because that effects the amount of encoding going on, which effects the extent to which memory will be expressed later. So I am left with the feeling of

being unsure about the veracity of the main effect. Perhaps a discussion of why these differences should not affect interpretation would help assuage these concerns.

Authors' response: A similar point has been raised by Reviewer #1 comment 4. The figure S1 exclusively shows data related to the retrieval phase. As suggested by both reviewers, we have now added respective results to the main text (and also extended the description of these data in the Supplemental material.) Please, note that the original fig. S1 is now fig. S2 to keep the order of figures according to their mentioning in the text. The changes to the main text are as follows (Results section, p. 4-6):

“...The enhancement was largest during the first minute of the retrieval phase which is typically most sensitive to the memory effect²⁴ ($F(2, 51) = 4.464, P = 0.017$, for main effect of Group, see Fig. 1B for results from pairwise statistical comparisons). In fact, in the 1st min of retrieval testing. Analysis of behavioral control parameters revealed that Spatial-experience and Object-experience groups were closely comparable with respect to total exploration time (fig. S2A). However, rats of the Spatial-experience group travelled a slightly greater distance than the rats of the Object-experience group ($t(32) = -2.055, P = 0.048$), possibly reflecting general arousing effects on locomotion resulting from stimulation specifically of spatial systems during infancy.

Control analyses of the behavior at the four exposures during infancy...”

Reviewer's comment 3: "Adult rats show no episodic memory for spatial exposures during infancy"

Here, the question of what is remembered is addressed, and the conclusion is that what is remembered is not an episodic memory for the ‘spatial experience’ or ‘spatial exposure’ (meaning the spatial environment, I guess), but something else. The evidence for this is that a single exposure to the sample locations – not the whole task done four times, as in the previous experiment – resulted in no evidence of memory for the object locations. The authors emphasize that this is the “strictest test” of episodic memory. But that means the way it’s tested biases the result toward the eventual conclusion. I can’t see how it tells us much about what is remembered in the previous experiment, as it doesn’t at all resemble the previous experiment. Truth is, I’m not sure exactly why this experiment was done.

And in fact, some evidence was found for intact memory (significant preference for the novel location, which indicates memory). However this is interpreted as a “rudimentary” type of memory, rather than episodic. No reason is given for this interpretation.

So at the end of this experiment, we don’t yet know what is and is not remembered in the initial main experiment. At this point there are still no findings that are shown to be specific to infantile amnesia.

Authors' response: We thank the reviewer for bringing this up. Yes, we agree with the Reviewer that this experiment is not at the core of our study, and we also agree that with these experiments no findings are presented that are “specific to infantile amnesia”. We, therefore, summarized results of these experiments in a new figure (fig. S5) which we moved to the Supplementary material. In the text we clarified the rationale for this experiment – i.e., to exclude that the OPR performance enhancement at adulthood testing as observed in the Spatial-

experience group, could be basically produced by an episodic memory for an individual episode (exposure) experienced during infancy. Results of this experiment are negative – the rats tested at adulthood do not express the standard behavioral signs of an episodic memory for the exposure. In this regard, these findings are consistent with the phenomenon of “infantile amnesia”. However, please note, the main goal of our study was not to characterize infantile amnesia, i.e., what of an episode experienced during infancy is forgotten but rather, what of an infantile experience is maintained and kept in memory into adulthood.

We clarified the rationale for this experiment as follows (p. 9):

“In a further control experiment, we asked whether the enhanced spatial memory capabilities in the Spatial-experience group were perhaps a direct consequence of an episodic memory that was formed for the individual exposures of the spatial experience during infancy and persisted into adulthood. To answer this question...”

Figure S5 summarizes the findings of this experiment:

Figure S5. OPR memory for the Long-term OPR control group. Findings are shown in comparison with the Spatial-experience group of the main experiments. (A) General procedure for both groups. For the Long-term OPR ($n = 12$ rats, cyan), the OPR encoding phase took place during infancy (PD24) and retrieval testing at adulthood (PD84). (B) Mean \pm SEM

discrimination ratios shown separately for the first 1, 2, 3, 4 and entire 5 min of the retrieval phase, and (C) total object exploration time (in s; left) and total distance travelled (in m) during the 1st min of retrieval phase for the Long-term OPR (cyan bars) and Spatial-experience groups (red bars; dot plots overlaid) # $P < 0.05$, ## $P < 0.01$ and ### $P < 0.001$ for one-sample t-test against chance level. * $P < 0.05$, ** $P < 0.01$ and *** $P < 0.001$, for pairwise comparisons (two-sided t-tests) between experimental groups. In (B), discrimination ratios differed between groups across the entire 5-min phase, $F(1, 27) = 7.490$, $P = 0.011$, for group main. Note, starting from the 2nd minute, the Long-term OPR group displayed significant negative discrimination ratios suggesting the presence of a rudimentary form of a memory for the original infantile experience which, curiously, is expressed in an “infantile” manner, namely as familiarity preference (for the stationary object) rather than as novelty preference²¹. In (C), the diminished total exploration time and distance travelled in the Long-term OPR group partly reflect the use of a smaller arena for retrieval testing (because for this group, the encoding phase took place during infancy where a smaller arena was used). The difference in OPR memory between the groups was confirmed in an analysis of covariance on the discrimination index using total exploration time and distance travelled as covariates ($F(1, 28) = 4.570$, $P = 0.042$ and $F(1,28) = 11.114$, $P = 0.003$, respectively).

We changed the last (summarizing) paragraph to further clarify the main goals of our study (p. 24-25):

“... supraordinate contextual memory in cortical networks^{12, 39}. Sleep has been shown to be critical for the quick transformation of experience encoded in the hippocampus-dependent episodic memory system, into less detailed representations mainly residing in cortical networks⁶³, and such transformation might comprise the simultaneous forgetting of the episodic, presumably hippocampal, representation. In this view, sleep might also contribute to the forgetting of episodes experienced during infancy, as it was observed in the rats of our Long-term OPR group (fig S5) and as it constitutes the phenomenon of infantile amnesia^{35, 64, 65}. However, rather than infantile amnesia and the forgetting of episodic memory during infancy, the central question addressed by our study was about the information experienced in early-life that is *not* forgotten but maintained in memory for the long term. In this regard, our findings suggest that, mediated by a sleep-dependent mechanism, the infant brain preferentially forms long-term memories for contextual information which critically involve mPFC networks...”

Reviewer’s comment 4: "Effects of infantile spatial experience on the adult rat’s learning capability are context-dependent".

The authors “hypothesized that infantile spatial experience improved adult spatial memory capabilities via inducing a more abstract schema-like representation which in theory is context-independent.” Thus, in this experiment two groups of rats were tested, one that had OPR task exposure in context 1, and OPR testing in context 2, and another that had OPR task and testing in the same context (the rats in the first experiment). The manipulation of context extended to the identity of the experimenter, which is good and thorough. Only when the context matched did rats benefit from pre-exposure to the task. This is what one would expect, based on the idea (see above) that it is encoding of the spatial environment (context) that helps the pre-exposed animals perform better. The authors conclude “These results support the notion that schema-

like spatial memories formed during infancy can also integrate contextual elements of the infantile experience.” This is what would be predicted, and so doesn’t really say much new, except for the statement that this contextual/spatial memory is “schema-like” (in the Discussion it is described as “abstract”). Unfortunately, so far in these experiments I don’t see any evidence that the memories are schema-like or abstract. To test that, experiments would need to be specifically designed to test whether memories are “schema-like” or not. Indeed, the hypothesis above says that a schema-like representation would be context-independent. But the facilitation by pre-exposure is context-dependent, and so isn’t the conclusion the opposite, that the memory is not schema-like? I am getting a little confused at this point.

At this point there are still no findings that are shown to be specific to infantile amnesia.

Authors’ answer: Yes, this comment is related to this Reviewer’s comment 1 which we addressed by adding a “Stationary-experience” group (see above). The findings from these experiments suggested that improvement in OPR performance at adulthood in the Spatial-experience group is most likely based on the formation of memory for the environmental arena context during the infantile experience. If so, the improvements in OPR performance at adulthood in the Spatial-experience group would be expected to be context dependent. In fact, this is what the findings of these experiments showed. We have changed the rationale for these experiments accordingly (please, see the respective changes to the text in our response to this Reviewer’s comment 1) and have completely abandoned the terms “schema”, “schema-like” or “abstracted” in conjunction with the respective memories.

Reviewer’s comment 5: "Infantile spatial experience forms memory in the prelimbic region of the mPFC"

In this experiment, the authors test whether whatever-memory-is-responsible-for-improved-performance-in-the-S-E-group is transferred from hippocampus to prefrontal cortex in a systems-level-consolidation-type manner. This is done using c-fos imaging, comparing activity in hippocampus and prefrontal cortex. First it needs to be acknowledged that the results will be hard to interpret because the comparison between groups is confounded by performance: the S-E groups perform the task well prior to c-fos analysis, and the other groups don’t. Putting that aside for the moment, no between-group differences in c-fos were found in the hippocampus. Instead, differences were found in the prefrontal cortex, specifically in the prelimbic cortex where c-fos activity in S-E group was greater than in the other 2 groups. This agrees with the expectations of the study. However this finding was accompanied by a number of other unexpected (I think) changes such as greater activity in the PAR, PRC, and LEC in the No-exposure group compared to the S-E group. I have no good explanation for these unexpected findings, but it does make me wonder whether we should be focussing and putting our trust in the one finding that agrees with the expectations of the study, when there are all these other changes we don’t understand. Having said that, the prelimbic finding does seem sound as it was subsequently replicated in other ways.

As mentioned above, there has been no convincing evidence presented so far that the facilitating memories are schema-like, or abstract, or non-episodic. Throughout this section a lot of conclusions are made about the nature of the memory, based on the structures active according to c-fos etc. But structures are not memories, and although the system-level-consolidation episodic-to-schema type model is believed by many, it is also heavily contested. So we really can’t be making conclusions about the nature of a memory on the basis of the

activity of brain structures, even if the data were clear, unconfounded, and unproblematic. And at this point there are still no findings that are shown to be specific to infantile amnesia. Also I'd prefer a causal experiment to test the necessity of the putative brain structures. We get that next.

Authors' answer: Yes, we agree with the Reviewer that a “correlational” approach, i.e., mapping OPR performance onto increases in c-Fos activity in specific brain regions, itself remains inconclusive and can be confounded in many ways. As suggested, we have now explicitly acknowledged that “the results are hard to interpret because the comparison between groups is confounded by performance”. We also agree with the Reviewer that valid conclusions as to the contribution of specific brain regions to the target representations can only be drawn in conjunction with “a causal experiment” (which we did by suppressing activity in the prelimbic mPFC region during OPR performance, see below).

We have acknowledged the possible confound of the c-Fos findings by differences in performance in the text by adding it to the rationale to our “causal experiments” (p. 13-14):

“Inhibiting the prelimbic region of mPFC at adult OPR testing abolishes benefits from early spatial experience

Our examination of c-Fos activity indicated increased activity at OPR testing during adulthood specifically in the PL region of the mPFC in the Spatial-experience group in comparison with both the Object-experience and No-experience control groups. The mere association of increased c-Fos with enhanced OPR performance, however, can be confounded by various factors (e.g., the differences in overt behavior between the groups) and does not allow for valid conclusions as to the contribution of a specific brain region to the memory of interest. Therefore, to test whether representations in the PL region of the mPFC causally contribute to mediating the enhancing effects on spatial learning during adulthood, two further groups of rats were added which were subjected to the same protocol as the Spatial experience group of the main experiment, with four exposures to changes in spatial configurations, However, ...”

Reviewer's comment 6: "Inhibiting the prelimbic region of mPFC at adult OPR testing abolishes benefits from early spatial experience".

Prelimbic cortex was silenced prior to the retrieval phase of the OPR task, during the test (not the pre-exposure). Performance was impaired in the muscimol group compared to the saline group (both groups had spatial pre-exposure). It is concluded that “These results indicate that the PL mPFC is critically involved in mediating the improving effects of infantile spatial experiences on the adult rat's spatial learning capabilities.” However the effect could be just muscimol impairing the task, rather than anything to do with the effects of pre-exposure. A control for this might be something like muscimol and saline in rats that performed the task well, but that had not been pre-exposed.

And at this point there are still no findings that are shown to be specific to infantile amnesia. But wait ...

Authors' response: We thank the reviewer for this important comment. To exclude that the suppression of the PL mPFC basically and independently of the prior infantile experience

interferes with animal’s capability to successfully perform the OPR task, we performed further control experiments in a new group of naive adult rats, employing the same procedures as in the in the main muscimol/saline experiments except that the retention interval between the encoding and retrieval phase was shortened to 18 min, i.e., conditions in which adult rats typically display successful OPR retrieval. In these experiments, the rats indeed displayed significant and comparable OPR memory in both the muscimol and vehicle conditions, thus excluding that muscimol interfered with the rat’s capability to perform the task. We have integrated results from these experiments in the main text as follows (p. 14-16):

“...Total exploration time and total distance travelled at retrieval testing was comparable between groups ($F(1,40) = 1.08$ for distance travelled, $F(1,40) = 2.63$ for total exploration time, $P > 0.1$, one-way ANOVA).

Although these results support the view of the PL region of the mPFC being critically involved in mediating the improving effects of infantile spatial experience on OPR performance in adulthood, it could be argued that, in these experiments, the suppression of the PL mPFC by muscimol basically and independently of any prior infantile experience interfered with OPR performance. To exclude this possibility we performed additional control experiments in a new group of naïve adult rats (~PD80) using the same procedures as in the main experiments except that the OPR retention interval between encoding and retrieval testing was shortened to 18 min (i.e., conditions where adult rats normally show significant OPR memory). In these experiments, the rats showed significant OPR memory after infusion of vehicle as well as after muscimol ($t(23) > 2.456$, $P < 0.022$, for both conditions, with no difference between conditions, $P = 0.41$, fig. S11).

In combination, results from these experiments indicate that the PL mPFC plays a causal role in mediating the improving effects of infantile spatial experiences on the adult rat’s spatial learning capabilities. In conjunction ...”

Figure S11 summarizes results from these control experiments:

Figure S11. Inhibition of the PL mPFC by muscimol does not impair adult rat's capability for OPR memory. (A) Procedures were the same as in the main experiments (Fig. 4) except that the retention interval between encoding and retrieval was shortened to ~18 min, and there was no prior infantile experience. Muscimol/saline was infused immediately after the encoding phase bilaterally over 3 min and 15 min later, the OPR retrieval phase started. Each of 4 rats was tested twice on the muscimol and vehicle conditions, respectively, in random order (using different objects and configurations at each test and a >2 days interval between conditions). (B) Discrimination ratios and (C) total object exploration time (in s, top) and distance travelled (in m, bottom). Mean±SEM values across first 3 min of retrieval phase are shown, dot plots overlaid (n = 8). # $P < 0.05$, ## $P < 0.01$, for one sampled t-test against chance level. Note, significant OPR memory in both muscimol and vehicle conditions. There were no significant differences between conditions (all $P > 0.41$).

Reviewer's comment 7: "Effects of early spatial experience on adult OPR performance depend on developmental age"

Here spatial exposure is given to rats at three different ages -- infancy, early childhood and adolescence – and OPR tested in adulthood. The infancy group performed well. Rats exposed in early childhood or adolescence did not perform well, suggesting that the initial finding was specific to infantile amnesia. It's great that an experiment was included to test whether the main finding was specific to infantile amnesia. However some features of the data make me uneasy. For example, the Infancy group has an N of 27, and the Early Childhood group has an N of 7. Putting aside the statistical problem of uneven sample sizes, can we really trust the conclusions when they depend on finding an effect in the highly-powered group, and no effect in the low-powered group?

In any case what I don't understand is why this would happen, even according to the authors' favored interpretation. The story here is that the infants don't form episodic memories, only "rudimentary", "abstract" ones. The older groups would have episodic memories of the experience. So why does "rudimentary" memory facilitate subsequent performance, but not episodic memory? This is counter-intuitive, and needs to be explained.

Authors' response: Many thanks for this interesting comment! Yes, in theory, the early childhood and adolescent rat groups are expected to form episode memories of the spatial experience, at least to a greater extent than the infantile group. Why didn't they benefit from this spatial experience to the same or even greater extent than the infantile group? This is difficult to answer, even in light of the new findings indicating that the benefit in the infantile group relied on context memory (see our response to comment #1 by this Reviewer). One possibility is that episodic memory formation in the relatively older juvenile rats prevented or interfered with the selective consolidation of context into mPFC networks as observed for the infantile group. Also, mechanisms of critical period plasticity may play a role, as suggested e.g., by C. Alberini's group (Travaglia et. al., 2016). However, such explanation remain highly speculative (also because we did not include standard tests of episodic memory at early childhood and adolescence).

As to the statistical aspects of the comment, we want to clarify that the early childhood group includes 11 (not 7) rats. The infantile group and the combined early childhood and adolescent groups thus end up with roughly comparable size (n = 27 vs n = 11 + 17). Pooling

the two older groups is, in this case, a statistically legitimate procedure, although we agree that it can be questioned on conceptual grounds. Against this background, and also because we do not see infantile amnesia in terms of forgetting of episodic experience in the center of our study (please, see our response to comment #3), we decided not to extend the discussion of the data here, but to more explicitly caution against drawing any premature conclusions from these data. Accordingly, the text (p. 18) was changed as follows:

“... groups; Fig. 5A). Overall, the pattern of results agrees with our hypothesis that infancy is a period of particular sensitivity to spatial experiences and for taking them to build contextual long-term memories⁷. However, in light of the relatively small group sizes these conclusions remain tentative. “

Reviewer’s comment 8: "Effects of early spatial experience on adult OPR memory depend on post-experience sleep"

In this experiment, sleep deprivation in infancy affected performance on test in adulthood. But we don’t know from this experiment whether sleep deprivation affected the facilitatory effects of pre-exposure, or just performance of the task in adulthood.

Authors’ response: A very similar comment regarding unspecific side effects of the sleep deprivation procedure (particularly stress) has been made by Reviewer #1 (comment 5). Yes, although unlikely, we can presently not entirely exclude that deprivation of the pups induced stress that confounded the effects of mere wakefulness and might have induced more general adverse effects on the pup’s brain development. We have extended the discussion of this issue as follows (p. 19):

“...experience group ($t(10) = 2.768$, $P = 0.020$, for pairwise comparison between groups).

It is unlikely that these effects of sleep deprivation were confounded by stress capable to induce general adverse effects on brain development, as sleep deprivation of the pups was performed very cautiously using gentle handling in the presence of the littermates and close to the mother that could be seen and smelt by the pup. Moreover, duration of sleep deprivation was relatively short (1.5 h), and none of the pups showed any behavioral signs of stress or fear during sleep deprivation. The use of similar deprivation procedures in adult rats remain without any effect on stress hormones levels like corticosterone (e.g., Melo and Ehrlich, 2016). Thus, assuming that the sleep deprivation procedure did not induce substantial side effects, our findings support the conclusion that the beneficial effect of infant spatial experience on adult spatial learning requires sleep to occur after the infant experience ...”

Minor comments.

Reviewer’s minor comment 1: In the Introduction, it is made clear that a number of studies have already looked at this problem and have offered some answers. That suggests that perhaps this study is not so novel. So what is new? What is new, according to the Introduction, is that previous studies all used aversive methodology to study ‘emotional’ memories, whereas as the present study does not. To the non-expert, this might seem like a possibly trivial tweak in methodology. Rather than just state that something was done differently, I think it needs to be

explained why the shift from aversive to non-aversive methodology is important. How might the previous studies have given us the wrong answer because of the methodology used? Is it due to possible confounds, for example the effects of stress and stress hormones? I think the rationale and importance of this novel aspect of the study could be expanded and made more explicit.

Authors' response: Many thanks for this comment! We have now more clearly expressed, in the Introduction, the importance to “shift from aversive to non-aversive methodology” as follows (Introduction, p. 3-4):

“... Despite the outstanding theoretical interest, little experimental work has been performed on how a certain separate episodic experience during infancy is consolidated into memory to feed into adult knowledge systems^{11, 13-18}. Moreover, most of this research employed highly aversive stimuli that, due to their stressful nature, invoke memory processes distinctly differing from more neutral everyday experience (Schwabe et al. 2022; Miranda et al., 2022; Travaglia et al., 2016), and the very few studies examining effects of non-aversive experience all employed non-specific stimulation covering extended periods of postnatal life (like prolonged exposure to novel environments²¹⁻²³). Against this backdrop, in the present study we adopted a novel experimental approach to tackle the question how discrete non-emotional spatial experience during infancy becomes integrated into persisting knowledge systems and eventually impacts adult behavior. We took advantage of the well-controlled conditions of a rat model to show that such discrete experiences, i.e., the exposure to a change in the spatial configuration of two objects in an experimental arena for only short 5-min intervals on 4 days during a rat's infancy, distinctly enhances the rat's spatial learning ability during adulthood. To induce...”

Reviewer's minor comment 2: The data in Fig 1B show that only the S-E group performed well on the OPR task; the other groups attained scores that are for the most part no different from chance. The authors say this low performance “replicates previous studies” but cite only one study from their own lab. These tasks have been used in a huge number of studies, and performance this poor is highly unusual. This does not negate the fact that the S-E group did perform well, and indeed one might have intentionally designed the study using parameters that generated low baselines so that improvements could be more easily seen (for example, by using a 24-hour delay, or more similar locations). Nevertheless, the rats' performance is so poor at a 3-hour delay that I can't help but be concerned about how the task is being run. This concern is amplified by the huge amount of variability in the data; the whole range of possible scores is represented, from -1 to 1. This is also unusual for this type of task when run properly. In any case, I think readers will want to be convinced that the methodology is sound and so this issue should be discussed and cited more openly and accurately, and some possible reasons for the poor performance should ideally be given.

Authors' response: We thank the reviewer for bringing up this point allowing us to provide a elaborated rationale for the design of our OPR task used at adulthood. Indeed, as the Reviewer suspected, we intentionally designed the task such that baseline performance was expected to be low and more variable than usual. (We have used the identical task in a forgoing study - Contreras et. al., 2019 – with closely matching results). Importantly, we used a rather long 3-

hour retention interval between encoding and test. Although widely employed, to the best of our knowledge there are only a few studies showing significant OPR memory in rats with retention intervals of 3 hours or longer (Ozawa et al., 2011; Miguez et al., 2016; Sawangjit et al., 2020), and all of these studies used extensive training procedures to enhance encoding. By contrast, in our task protocol the encoding period was shortened to 5 min (compared to the 10 min used in standard OPR protocols). Moreover, an open field was used with only a few cues inside the arena on the walls so that animals for navigating could only rely on distal cues outside the arena.

We added this rationale for the OPR task design to the Methods (p. 28):

“At adulthood, all groups were tested on an Object-place recognition (OPR) task. This task which had been employed in an identical manner in a forgoing study (Contreras et. al., 2019) was designed such that recognition performance under baseline conditions was expected to be low. Specifically, to make the task more challenging, a long 3-hour delay between encoding and test was used in combination with a rather short 5-min encoding period. Encoding difficulty was further enhanced by the use of an open field with only a few cues inside the arena enforcing the animal to navigate based on distal cues outside the arena (see section Apparatus and objects). During the 3-hour retention interval the rats were moved to the home cage. Preparation for OPR testing included exactly...”

Reviewer’s minor comment 3: The citations are often focussed on a few currently popular authors (newcomers to this area, as well) and reviews in ‘glamor’ publications. So this is just a reminder to the authors that we always want to cite the earliest, primary sources.

Authors’ response: Yes, we have added “earliest, primary” references on several occasions, e.g., (p. 3)

“...little experimental work has been performed on how a certain separate episodic experience during infancy is consolidated into memory to feed into adult knowledge systems^{11, 13-16} (Bauer and Dow, 1994; Bauer et. al., 1998)”

(p. 16)

“... The first years of life are characterized by distinct conditions of synaptic plasticity and the shaping of neuronal circuits mediating memory formation, and many of these conditions, like a strongly elevated neurogenesis, particularly apply to the hippocampus, i.e., the structure centrally involved in the formation of spatial representations^{14, 32, 33} (Jabès et. al., 2011)...”

(p. 22)

“...Indeed, the conditions are reminiscent to those found in 15-20 months old human infants who did not discriminate the spatial room context while searching for toys hidden in boxes but, surprisingly, showed the ability to disambiguate the boxes according to the room context when the toy cues were absent (Newcombe et. al., 2014). Nevertheless, ...”

References

- Melo, I., Ehrlich, I. Sleep supports cued fear extinction memory consolidation independent of circadian phase. *Neurobiol. Learn. Mem.* **132**, 9-7 (2016).
- Contreras, M.P., Born, J., Inostroza, M. The expression of allocentric object-place recognition memory during development. *Behav. Brain Res.* **372**, 112013 (2019).
- Newcombe, N.S., Balcomb, F., Ferrara, K., Hansen, M., Koski, J. Two rooms, two representations? Episodic-like memory in toddlers and preschoolers. *Dev. Sci.* **17**(5), 743-56. (2014).
- Ekstrom, A.D., Ranganath, C. Space, time, and episodic memory: The hippocampus is all over the cognitive map. *Hippocampus.* **28**, 680-687 (2018).
- Burgess, N., Maguire, E.A., O'Keefe, J. The human hippocampus and spatial and episodic memory. *Neuron* **35**, 625-641 (2002).
- Robin, J., Moscovitch, M. Familiar real-world spatial cues provide memory benefits in older and younger adults. *Psychol. Aging.* **32**, 210-219 (2017).
- Hassabis, D., Maguire, E.A. Deconstructing episodic memory with construction. *Trends. Cogn. Sci.* **11**, 299-306 (2007).
- Travaglia, A., Bisaz, R., Sweet, E.S., Blitzer, R.D., Alberini, C.M. Infantile amnesia reflects a developmental critical period for hippocampal learning. *Nat. Neurosci.* **19**, 1225 (2016).
- Schwabe, L., Hermans, E.J., Joëls, M., Roozendaal, B. Mechanisms of memory under stress. *Neuron* **110**, 1450-1467 (2022).
- Miranda, J.M., Cruz, E., Bessières, B., Alberini, C.M. Hippocampal parvalbumin interneurons play a critical role in memory development. *Cell Rep.* **4**, 111643 (2022).
- Ozawa, T., Yamada, K., Ichitani, Y. Long-term object location memory in rats: effects of sample phase and delay length in spontaneous place recognition test. *Neurosci. Lett.* **497**, 37-41 (2011).
- Migues, P.V., Liu, L., Archbold, G.E., Einarsson, E.Ö., Wong, J., Bonasia, K., Ko, S., Wang, Y.T., Hardt O. Blocking Synaptic Removal of GluA2-Containing AMPA Receptors Prevents the Natural Forgetting of Long-Term Memories. *J. Neurosci.* **36**, 3481-3494 (2016).
- Sawangjit, A., Oyanedel, C.N., Niethard, N., Salazar, C., Born, J., Inostroza, M. The hippocampus is crucial for forming non-hippocampal long-term memory during sleep. *Nature* **564**, 109-113 (2018).

Bauer, P.J., Dow, G.A. Episodic Memory in 16- and 20-Month-Old Children: Specifics Are Generalized but Not Forgotten. *Dev. Psychol.* **30**, 403-417 (1994).

Bauer, P.J., Dow, G.A., Bittinger, K.A., Wenner, J.A. Accepting and exempting the unexpected: 30-month-olds' generalization of event knowledge. *Cog. Dev.* **13**, 421-452 (1998).

Jabès, A., Lavenex, P.B., Amaral, D.G., Lavenex, P. Postnatal development of the hippocampal formation: a stereological study in macaque monkeys. *J. Comp. Neurol.* **519**(6), 1051–1070 (2011).

REVIEWER COMMENTS

Reviewer #1 (Remarks to the Author):

I am happy to recommend this revised manuscript for publication.

Reviewer #3 (Remarks to the Author):

I think the authors have done a really great job of taking on the reviewer's comments. There is one thing I still think they need to do, and I apologize to the authors for not being clearer in my initial review. In my "Comment 2" I said

"Even more important [to analyze] is the exploration during pre-exposure, because that effects the amount of encoding going on, which effects the extent to which memory will be expressed later."

What I meant by "pre-exposure" was the sample phase, or what the authors call the encoding phase. "Pre-exposure" was a very poor choice of words, especially since it means something else in the context of this study, so I apologize. In any case it is really important in these types of experiments to analyze sample exploration, because the time a subject spends exploring an object/location impacts the extent to which it is encoded, and therefore how well it will be remembered. Therefore if a manipulation affects the amount of sample exploration, this can result in a failure at retrieval that will mistakenly be attributed to the manipulation.

The authors have already analyzed overall exploration, and I doubt the sample exploration analysis will reveal much different, but it needs to be done, in any experiment using the spontaneous object/location recognition paradigm.

I should be clear it needs only to be done for the test (e.g., at adulthood), not during e.g., pre-exposure (e.g., infancy). The muscimol experiment probably doesn't require it because the manipulation is done following the sample phase. (But it probably should be done anyway for consistency.)

A final comment is about the control experiment added to the muscimol/PFC experiment (my Comment 6). I suggested a control "something like muscimol and saline in rats that performed the task well, but that had not been pre-exposed." This is exactly what the authors did, but what I hadn't thought enough about was how to make the task easier so that the non-pre-exposed animals would perform well. The way the authors did it was to shorten the delay. But now, the control test has a lower memory requirement, and that alone could explain why the muscimol had less of an effect in the adults.

The fact is I don't know what to do about this. But what the authors did is a lot better than nothing!

Point-by-point responses to the remaining comments by Reviewer #2

We very much appreciate the Reviewer's overall positive evaluation of our work, and we are happy to address his/her remaining two points which we, again, found very helpful to further improve our manuscript.

Reviewer's comment 1: In my "Comment 2" I said: "Even more important [to analyze] is the exploration during pre-exposure, because that effects the amount of encoding going on, which effects the extent to which memory will be expressed later."

What I meant by "pre-exposure" was the sample phase, or what the authors call the encoding phase. "Pre-exposure" was a very poor choice of words, especially since it means something else in the context of this study, so I apologize. In any case it is really important in these types of experiments to analyze sample exploration, because the time a subject spends exploring an object/location impacts the extent to which it is encoded, and therefore how well it will be remembered. Therefore if a manipulation affects the amount of sample exploration, this can result in a failure at retrieval that will mistakenly be attributed to the manipulation.

The authors have already analyzed overall exploration, and I doubt the sample exploration analysis will reveal much different, but it needs to be done, in any experiment using the spontaneous object/location recognition paradigm.

I should be clear it needs only to be done for the test (e.g., at adulthood), not during e.g., pre-exposure (e.g., infancy). The muscimol experiment probably doesn't require it because the manipulation is done following the sample phase. (But it probably should be done anyway for consistency.)

Authors' response: We now added the total object exploration and distance travelled during encoding phase of the adult OPR test corresponding to the main groups of Fig. 1 in fig. S2B and S2D. Similarly, we included control variables during encoding phase of the groups under muscimol/saline conditions in fig. S10B and fig. S11D.

Please see below the changes made to the manuscript (main text and supplementary figures) as follows:

(next page)

Figure S2. Total exploration time and distance travelled during OPR performance. (A) Total object exploration time (in s) towards both objects (left) and total distance travelled (in m) during

the retrieval phase (1st minute) for the Spatial-experience (red bars, $n = 17$), Object-experience (green bars, $n = 17$) and No-experience groups (grey bars, $n = 18$). **(B)** The same control variables for the 5-min encoding phase in these groups. **(C)** Total object exploration time and total distance travelled for the control experiments comparing the Spatial-experience group (red bars) with the Stationary-experience ($n = 12$ rats, yellow) and the Context-change group ($n = 12$ rats, purple). **(D)** The same control variables for the 5-min encoding phase in these groups. Means (\pm SEM) are indicated. * $P < 0.05$ and *** $P < 0.001$ for pairwise t -test (two-sided). Note, in **(A)** Spatial-experience and Object-experience groups were closely comparable with respect to total exploration time. However, the Spatial-experience group travelled a slightly greater distance than the Object-experience group ($t(32) = -2.055$, $P = 0.048$), possibly reflecting general arousing effects on locomotion resulting from stimulation specifically of spatial systems during infancy. In **(B)**, both Spatial-experience and Object-experience groups showed increased total object exploration and distance travelled in the encoding phase in comparison with the No-experience control group ($t(33) > 3.6$, $P > 0.001$), suggesting an unspecific effect of infantile experience (independent of the kind of experience). To further exclude any confounding influence of these control variables that differed between groups on OPR retrieval performance, we ran additional ANOVA including these variables as covariate (i.e., distance travelled at retrieval or encoding, and total exploration at encoding). These analyses confirmed the significant differences in the discrimination index between groups (as reported in the main text) in all cases (all $P < 0.034$), with none of the covariates reaching significance (all $P > 0.153$). In **(D)** the Context-change group showed reduced total exploration time for the objects (when compared to the Spatial-experience group, $t(27) = 2.864$, $P = 0.015$) during retrieval, possibly due to increased context exploration. Additional ANOVA including total exploration time as covariate confirmed a significantly higher discrimination index in the Spatial-experience than the Context-change group ($P = 0.034$) thereby excluding that the differences in OPR memory were confounded by total exploration time ($P = 0.409$ for the covariate).

Results – In the main text, we referred to the behavioral control variables during the encoding phase as follows (p. 6):

“...Analysis of behavioral control parameters during the retrieval phase revealed that Spatial-experience and Object-experience groups were closely comparable with respect to total exploration time (fig. S2A). However, $P = 0.007$). During the encoding phase, total object exploration and distance travelled were decreased in the No-experience group ($t(33) > 3.6$, $P > 0.001$, for comparisons with the two other groups). Control analyses using total exploration and distance travelled at encoding as covariates did not provide any hint that these parameters biased the observed differences in the discrimination index between groups ($P > 0.153$ for the covariates, fig. S2).....”

For consistency, we have now also added total exploration and distance travelled during encoding for the muscimol experiments to fig. S10 and S11:

Figure S10. Adult OPR performance for the entire 5-minute retrieval phase for the Saline and Muscimol groups. (A) Mean±SEM discrimination ratios separately during the first 1, 2, 3, 4 and entire 5 min of the retrieval phase at adulthood OPR testing, for the Saline ($n = 6$) and Muscimol ($n = 6$) groups (dot plots overlaid). # $P < 0.05$, one sample t -test against chance level. Differences between the groups for the first 3 min were significant ($F(1, 34) = 8.4, P < 0.01$, for main effect of Group, $P = 0.84$ for Group x Minutes interaction, see main text). (B) Mean±SEM total object exploration time (in s, left) and distance travelled (in m, right) during the 5-min encoding phase. There were no significant differences between conditions (all $P > 0.05$).

Figure S11. Inhibition of the PL mPFC by muscimol does not impair adult rat's capability for OPR memory. (A) Procedures were the same as in the main experiments (Fig. 4) except that the retention interval between encoding and retrieval was shortened to ~18 min, and there was no prior infantile experience. Muscimol/saline was infused immediately after the encoding phase bilaterally over 3 min and 15 min later, the OPR retrieval phase started. Each of 4 rats was tested twice on the muscimol and vehicle conditions, respectively, in random order (using different objects and configurations at each test and a >2 days interval between conditions). (B) Discrimination ratios. Mean \pm SEM values across first 3 min of retrieval phase are shown, dot plots overlaid ($n = 8$). (C) Total object exploration time (in s, left) and distance travelled (in m, right) during retrieval and (D) the 5-min encoding phase. # $P < 0.05$, ## $P < 0.01$, for one-sample t -test against chance level. Note, significant OPR memory in both muscimol and vehicle conditions. There were no significant differences between conditions (all $P > 0.41$).

Reviewer's comment 2: A final comment is about the control experiment added to the muscimol/PFC experiment (my Comment 6). I suggested a control "something like muscimol and saline in rats that performed the task well, but that had not been pre-exposed." This is exactly what the authors did, but what I hadn't thought enough about was how to make the task easier so that the non-pre-exposed animals would perform well. The way the authors did it was to shorten the delay. But now, the control test has a lower memory requirement, and that alone could explain why the muscimol had less of an effect in the adults. The fact is I don't know what to do about this. But what the authors did is a lot better than nothing!

Authors' response: Yes, we very much agree that, as every control, this one also has a weakness. We have changed the phrasing in the text to more clearly express this weak aspect of the control:

(p. 15)

“... To exclude this possibility, we performed additional control experiments in a new group of naïve adult rats (~PD80) using the same procedures as in the main experiments except that the OPR retention interval between encoding and retrieval testing was shortened to 18 min (which **inevitably lowered memory requirements** but represent conditions where adult rats normally show significant OPR memory). In these experiments....”